# Gradient-based Hyperparameter Optimization Over Long Horizons

**Paul Micaelli**
University of Edinburgh
{paul.micaelli}@ed.ac.uk

**Amos Storkey**
University of Edinburgh
{a.storkey}@ed.ac.uk

## Abstract

Gradient-based hyperparameter optimization has earned a widespread popularity in the context of few-shot meta-learning, but remains broadly impractical for tasks with long horizons (many gradient steps), due to memory scaling and gradient degradation issues. A common workaround is to learn hyperparameters online, but this introduces greediness which comes with a significant performance drop. We propose forward-mode differentiation with sharing (FDS), a simple and efficient algorithm which tackles memory scaling issues with forward-mode differentiation, and gradient degradation issues by sharing hyperparameters that are contiguous in time. We provide theoretical guarantees about the noise reduction properties of our algorithm, and demonstrate its efficiency empirically by differentiating through $\sim 10^4$ gradient steps of unrolled optimization. We consider large hyperparameter search ranges on CIFAR-10 where we significantly outperform greedy gradient-based alternatives, while achieving $\times 20$ speedups compared to the state-of-the-art black-box methods. Code is available at: `https://github.com/polo5/FDS`

## 1 Introduction

Deep neural networks have shown tremendous success on a wide range of applications, including classification [1], generative models [2], natural language processing [3] and speech recognition [4]. This success is in part due to effective optimizers such as SGD with momentum or Adam [5], which require carefully tuned hyperparameters for each application. In recent years, a long list of heuristics to tune such hyperparameters has been compiled by the deep learning community, including things like: how to best decay the learning rate [6], how to scale hyperparameters with the budget available [7], and how to scale learning rate with batch size [8]. Unfortunately these heuristics are often dataset specific and architecture dependent [9]. They also don't apply well to new optimizers [10], or new tools, like batch normalization which allows for larger learning rates [11].

With so many ways to choose hyperparameters, the deep learning community is at risk of adopting models based on how much effort went into tuning them, rather than their methodological insight. The field of hyperparameter optimization (HPO) aims to find hyperparameters that provide a good generalization performance automatically. Unfortunately, existing tools are rather unpopular for deep learning, largely owing to their computational cost. The method developed here is a gradient-based HPO approach; that is, it calculates hypergradients, i.e. the gradient of some validation loss with respect to each hyperparameter. Gradient-based HPO should be more efficient than black-box methods as the dimensionality of the hyperparameter space increases, since it is able to utilize gradient information rather than rely on trial and error. In practice however, learning hyperparameters with gradients has only been popular in few-shot learning tasks where the horizon is short, i.e. where the underlying task is solved with a few gradient steps. This is because long horizons cause hypergradient degradation, and incur a memory cost that makes reverse-mode differentiation prohibitive.

35th Conference on Neural Information Processing Systems (NeurIPS 2021).

Greedy gradient-based methods alleviate both of these issues by calculating local hypergradients based on intermediate validation losses. Unfortunately, this introduces some bias [15] and results in a significant performance drop, which we are able to quantify in this work. We make use of forward-mode differentiation, which has been shown to offer a memory cost constant with horizon size. However, previous forward-mode methods don't address gradient degradation explicitly and are thus limited to the greedy setting [16, 17].

We introduce FDS (Foward-mode Differentiation with hyperparameter Sharing), which to the best of our knowledge demonstrates for the first time that hyperparameters can be differentiated non-greedily over long horizons. Specifically, we make the following contributions: **(1)** we propose to share hyperparameters through time, both motivating it theoretically and with various experiments, **(2)** we combine the above in a forward-mode differentiation algorithm, and **(3)** we show that our method can significantly outperform various HPO algorithms, for instance when learning the hyperparameters of the SGD-momentum optimizer.

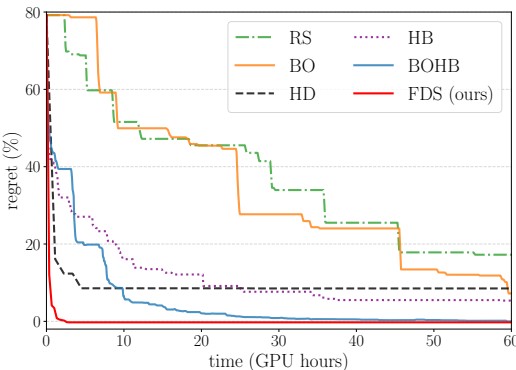

Figure 1: Performance of the most popular hyperparameter optimization methods when learning the learning rate, momentum and weight decay on 50 epochs of CIFAR-10, for a WideResNet of 16 layers. Non-greedy methods (RS, BO, HB [12], BOHB [13]) solve for global hyperparameters but rely on trial-and-error which makes them slow. Greedy gradient-based methods (e.g. HD [14]) are faster but solve for local hyperparameters which makes them suboptimal. Our method combines the strengths of these two paradigms and outperforms the next best method while converging 20 times faster. Each curve is the average of 8 seeds.

## 2  Related Work

There are many ways to perform hyperparameter optimization (HPO) [18], including Bayesian optimization (BO) [19], reinforcement learning [20], evolutionary algorithms [21] and gradient-based methods [22]. The state-of-the-art in HPO depends on the problem setting, but black-box methods like Hyperband (HB) [12], and its combination with BO into a method called BOHB [13] have been the most popular. Modern work in meta-learning deals with various forms of gradient-based HPO [23], but usually focuses on the few-shot regime [24] where horizons are conveniently short ($\sim 10$ steps) while we focus on long horizons ($\sim 10^4$ steps).

**Gradient-based HPO.**  Using the gradient of some validation loss with respect to the hyperparameters is typically the preferred choice when the underlying optimization is differentiable. This is a type of constrained optimization [25] which stems from earlier work on backpropagation through time [26] and real-time recurrent learning [27]. Unfortunately, differentiating optimization is an expensive procedure in both time and memory, and most proposed methods are limited to small models and toy datasets [28–30]. Efforts to make the problem more tractable include optimization shortcuts [31], truncation [32] and implicit gradients [33–35]. Truncation can be combined with our approach but produces biased gradients [36], while implicit differentiation is only applicable to hyperparameters that define the training loss (e.g. augmentation) but not to hyperparameters that define how the training loss is minimized (e.g. optimizer hyperparameters). Forward-mode differentiation [27] boasts a memory cost constant with horizon size, but gradient degradation has restricted its use to the greedy setting [16, 17].

**Greedy Methods.**  One trick that prevents gradient degradation and significantly reduces compute and memory cost is to solve the bilevel optimization greedily. This has become the default trick in various long-horizon problems, including HPO over optimizers [14, 16, 17, 37], architecture search [38], dataset distillation [39] or curriculum learning [40]. Greediness refers to finding the best hyperparameters locally rather than globally. In practice, it involves splitting the inner optimization problem into smaller chunks (often just one batch), and solving for hyperparameters over these smaller horizons instead; often in an online fashion. We formalize greediness in Section 3.2. In this paper we expand upon previous observations [15] and take the view that greediness fundamentally solves

for the wrong objective. Instead, the focus of our paper is to extend forward-mode differentiation methods to the non-greedy setting.

**Gradient Degradation.** Gradient degradation of some scalar w.r.t a parameter is a broad issue that arises when that parameter influences the scalar in a chaotic fashion, such as through long chains of nonlinear mappings. This manifests itself in HPO as vanishing or exploding hypergradients, due to low or high curvature components of the validation loss surface. This leads to hypergradients that have a large variance, making differentiation through long horizons impractical. This is usually observed in the context of recurrent neural networks [41, 42], but also in reinforcement learning [43] and HPO [29]. Solutions like LSTMs [44] and gradient clipping [45] have been proposed, but are respectively inapplicable and insufficient to long-horizon HPO. Variational optimization [36] and preconditioning warp-layers [46] can mitigate gradient degradation, but these methods are expensive in memory and therefore are limited to small architectures and/or a few hundred inner steps. In comparison, we differentiate over $\sim 10^4$ inner steps for WideResNets [47].

## 3 Background

### 3.1 Problem statement

Consider a neural network with weights $\boldsymbol{\theta}$, trained to minimize a loss $\mathcal{L}_{\text{train}}$ over a dataset $\mathcal{D}_{\text{train}}$. This is done by taking $T$ steps with a gradient-based optimizer $\Phi$, which uses a collection of hyperparameters $\boldsymbol{\lambda} \in \mathbb{R}^T$. For clarity of notation, consider that $\Phi$ uses one hyperparameter per step, written $\boldsymbol{\lambda}_{[t]}$, where indices are shown in brackets to differentiate them from a variable evaluated at time $t$. We can explicitly write out this optimizer as $\Phi : \boldsymbol{\theta}_{t+1} = \Phi(\mathcal{L}_{\text{train}}(\boldsymbol{\theta}_t(\boldsymbol{\lambda}_{[1:t]}), \mathcal{D}_{\text{train}}), \boldsymbol{\lambda}_{[t+1]}) \ \forall t \in \{1, 2, \ldots, T\}$. Note that $\boldsymbol{\theta}_t$ is a function of $\boldsymbol{\lambda}_{[1:t]}$, and it follows that $\boldsymbol{\theta}_T = \boldsymbol{\theta}_T(\boldsymbol{\lambda}_{[1:T]}) = \boldsymbol{\theta}_T(\boldsymbol{\lambda})$.

We would like to find the hyperparameters $\boldsymbol{\lambda}^*$ such that the result, at time $T$, of the gradient process minimizing objective $\mathcal{L}_{\text{train}}$, also minimizes some generalization loss $\mathcal{L}_{\text{val}}$ on a validation dataset $\mathcal{D}_{\text{val}}$. This can be cast as the following constrained optimization:

$$\boldsymbol{\lambda}^* = \arg\min_{\boldsymbol{\lambda}} \mathcal{L}_{\text{val}}(\boldsymbol{\theta}_T(\boldsymbol{\lambda}), \mathcal{D}_{\text{val}}) \tag{1}$$

$$\text{subject to} \quad \boldsymbol{\theta}_{t+1} = \Phi(\mathcal{L}_{\text{train}}(\boldsymbol{\theta}_t(\boldsymbol{\lambda}_{[1:t]}), \mathcal{D}_{\text{train}}), \boldsymbol{\lambda}_{[t+1]}) \tag{2}$$

The inner loop in Eq 2 expresses a constraint on the outer loop in Eq 1. In gradient-based HPO, our task is to compute the hypergradient $d\mathcal{L}_{\text{val}}/d\boldsymbol{\lambda}$ and update $\boldsymbol{\lambda}$ accordingly. Note that some methods require $\boldsymbol{\theta}_T = \arg\min_{\boldsymbol{\theta}} \mathcal{L}_{\text{train}}(\boldsymbol{\theta}, \mathcal{D}_{\text{train}}, \boldsymbol{\lambda})$ with $\mathcal{L}_{\text{train}}$ being a function of $\boldsymbol{\lambda}$ explicitly [33–35], in which case the above becomes a bilevel optimization [48]. Our algorithm waives these requirements.

### 3.2 Greediness

Let $H$ be the horizon, which corresponds to the number of gradient steps taken in the inner loop (to optimize $\boldsymbol{\theta}$) before one gradient step is taken in the outer loop (to optimize $\boldsymbol{\lambda}$). When solving Eq 1 non-greedily we have $H = T$. However, most modern approaches are greedy [14, 16, 37–39], in that they rephrase the above problem into a sequence of several independent problems of smaller horizons, where $\boldsymbol{\lambda}^*_{[t:t+H]}$ is learned in the outer loop subject to an inner loop optimization from $\boldsymbol{\theta}_t$ to $\boldsymbol{\theta}_{t+H}$ with $H \ll T$. Online methods such as Hypergradient Descent (HD) [14] are an example of greedy differentiation, where $H = 1$ and $\boldsymbol{\lambda}_{[t]}$ is updated at time $t$. Instead, non-greedy algorithms like FDS only update $\boldsymbol{\lambda}_{[t]}$ at time $T$.

Greediness mitigates gradient degradation and lowers the memory cost of backpropagation through time (BPTT). However, greedy methods do not solve for the actual objective in Eq 1, but for some proxy objective which minimizes intermediate validation losses evaluated at $\boldsymbol{\theta}_H, \boldsymbol{\theta}_{2H}, \ldots$. The degree to which these two objectives overlap is unknown and problem dependent. In our experiments, we found that finding competitive hyperparameters with greedy methods often revolves around tricks like online learning with a very low outer learning rate combined with hand-tuned initial hyperparameter values, to manually prevent convergence to small values. But solving the greedy objective correctly leads to poor solutions, a special case of which was previously described as the "short-horizon bias" [15]. In FDS, we learn global hyperparameters in a non-greedy fashion, which means that the hypergradient $d\mathcal{L}_{\text{val}}/d\boldsymbol{\lambda}$ is only ever calculated at $\boldsymbol{\theta}_T$.

## 3.3 Forward-mode differentiation of modern optimizers

The vast majority of meta-learning applications use reverse-mode differentiation in the inner optimization problem (Eq 2) to optimize $\boldsymbol{\theta}$. However, the memory cost of BPTT, namely using reverse-mode differentiation for the outer optimization (Eq 1) is $\mathcal{O}(FH)$, where $F$ is the memory used by one forward pass through the network (weights plus activations). In the non-greedy setting where $H = T$, BPTT is extremely limiting: for large networks, only $T \sim 10$ gradient steps could be solved with modern GPUs, while problems like CIFAR-10 require $T \sim 10^4$. Instead, we make use of forward-mode differentiation, which scales in memory as $\mathcal{O}(DN)$, where $D$ is the number of weights and $N$ is the number of learnable hyperparameters. The additional scaling with $N$ is a limitation if we learn one hyperparameter per inner step ($N = T$). Sharing hyperparameters (section 4.1) mitigates gradient degradation, but also conveniently allows for smaller values of $N$.

For clarity of notation, we consider forward-mode hypergradients for the general case of using one hyperparameter per step, i.e. $\boldsymbol{\lambda} \in \mathbb{R}^T$. First, we use the chain rule to write $d\mathcal{L}_{\text{val}}/d\boldsymbol{\lambda} = (\partial\mathcal{L}_{\text{val}}/\partial\boldsymbol{\theta}_T)(d\boldsymbol{\theta}_T/d\boldsymbol{\lambda})$ where the direct gradient has been dropped since $\partial\mathcal{L}_{\text{val}}/\partial\boldsymbol{\lambda} = 0$ for optimizer hyperparameters. The first term on the RHS is trivial and can be obtained with reverse-mode differentiation as usual. The second term is more problematic because $\boldsymbol{\theta}_T = \boldsymbol{\theta}_T(\boldsymbol{\theta}_{T-1}(\boldsymbol{\theta}_{T-2}(...), \boldsymbol{\lambda}_{[T-1]}), \boldsymbol{\lambda}_{[T]})$. We use the chain rule again to calculate this term recursively:

$$\frac{d\boldsymbol{\theta}_t}{d\boldsymbol{\lambda}} = \frac{\partial\boldsymbol{\theta}_t}{\partial\boldsymbol{\theta}_{t-1}}\bigg|_{\boldsymbol{\lambda}} \frac{d\boldsymbol{\theta}_{t-1}}{d\boldsymbol{\lambda}} + \frac{\partial\boldsymbol{\theta}_t}{\partial\boldsymbol{\lambda}}\bigg|_{\boldsymbol{\theta}_{t-1}} \quad \text{which we write as} \quad \boldsymbol{Z}_t = \boldsymbol{A}_t\boldsymbol{Z}_{t-1} + \boldsymbol{B}_t \qquad (3)$$

where $\boldsymbol{\theta}_t \in \mathbb{R}^D$, $\boldsymbol{\lambda} \in \mathbb{R}^T$, $\boldsymbol{Z}_t \in \mathbb{R}^{D \times T}$, $\boldsymbol{A}_t \in \mathbb{R}^{D \times D}$ and $\boldsymbol{B}_t \in \mathbb{R}^{D \times T}$. Note that the columns of $\boldsymbol{Z}$ at step $t$ are zeros for indices $t+1, t+2, ..., T$, since the hyperparameters at those steps haven't been used yet.

The expressions for $\boldsymbol{A}_t$ and $\boldsymbol{B}_t$ depend on the specific hyperparameters we are differentiating. In this work, we consider the most popular optimizer, namely SGD with momentum and weight decay. To the best of our knowledge, previous work focuses on simpler versions of this optimizer, usually by removing momentum and weight decay, and only learns the learning rate, greedily. We use Pytorch's update rule for SGD [49], namely $\boldsymbol{\theta}_t = \Phi(\boldsymbol{\theta}_{t-1}) = \boldsymbol{\theta}_{t-1} - \alpha_t\boldsymbol{v}_t$ with learning rate $\alpha_t$, momentum $\beta_t$, weight decay $\xi_t$ and velocity $\boldsymbol{v}_t = \beta_t\boldsymbol{v}_{t-1} + (\partial\mathcal{L}_{\text{train}}/\partial\boldsymbol{\theta}_{t-1}) + \xi_t\boldsymbol{\theta}_{t-1}$. Consider the case when we learn the learning rate schedule, namely $\boldsymbol{\lambda} = \boldsymbol{\alpha}$. If we use the update rule without momentum [17], $\boldsymbol{B}_t$ is conveniently sparse: it is a $D \times T$ matrix that only has one non-zero column at index $t$ corresponding to $\partial\boldsymbol{\theta}_t/\partial\alpha_t$. However, we include terms like momentum and therefore the velocity depends on the hyperparameters of previous steps. In that case, a further recursive term $\boldsymbol{C}_t = (\partial\boldsymbol{v}_t/\partial\boldsymbol{\lambda})$ must be considered to get exact hypergradients. Putting it together (see Appendix A) we obtain:

$$\begin{cases} \boldsymbol{A}_t^{\boldsymbol{\alpha}} = \boldsymbol{1} - \alpha_t \left( \dfrac{\partial^2\mathcal{L}_{\text{train}}}{\partial\boldsymbol{\theta}_{t-1}^2} + \xi_t\boldsymbol{1} \right) \\[2ex] \boldsymbol{B}_t^{\boldsymbol{\alpha}} = -\beta_t\alpha_t\boldsymbol{C}_{t-1}^{\boldsymbol{\alpha}} - \delta_t^{\otimes}\left( \beta_t\boldsymbol{v}_{t-1} + \dfrac{\partial\mathcal{L}_{\text{train}}}{\partial\boldsymbol{\theta}_{t-1}} + \xi_t\boldsymbol{\theta}_{t-1} \right) \\[2ex] \boldsymbol{C}_t^{\boldsymbol{\alpha}} = \beta_t\boldsymbol{C}_{t-1}^{\boldsymbol{\alpha}} + \left( \xi_t\boldsymbol{1} + \dfrac{\partial^2\mathcal{L}_{\text{train}}}{\partial\boldsymbol{\theta}_{t-1}^2} \right) \boldsymbol{Z}_{t-1}^{\boldsymbol{\alpha}} \end{cases} \qquad (4)$$

where $\boldsymbol{1}$ is a $D \times D$ identity matrix, and $\delta_t^{\otimes}(\boldsymbol{q})$ turns a vector $\boldsymbol{q}$ of size $D$ into a matrix of size $D \times T$, whose $t$-th column is set to $\boldsymbol{q}$ and other columns to $\boldsymbol{0}$s. While the matrices in Eq 4 are updated online, the hyperparameters aren't. This is because in the non-greedy setting we don't have access to the hypergradients until we have computed $\boldsymbol{Z}_T^{\boldsymbol{\alpha}}$. A similar technique can be applied to momentum and weight decay to get $\boldsymbol{Z}_T^{\boldsymbol{\beta}}$ and $\boldsymbol{Z}_T^{\boldsymbol{\xi}}$ (see Appendix A). All hypergradient derivations in this paper were checked with finite differences. Note that we focus on learning the hyperparameters of SGD with momentum because it is the most common optimizer in the deep learning literature. However, just like reverse-mode differentiation, forward-mode differentiation can be applied to any differentiable hyperparameter, albeit with appropriate $\boldsymbol{A}_t$ and $\boldsymbol{B}_t$ matrices.

# 4 Non-greedy Differentiation Over Long Horizons

## 4.1 Hyperparameter sharing: trading off noise reduction with bias increase

The main challenge of doing non-greedy meta-learning over long horizons is gradient degradation. In HPO this arises because a small change in $\mathcal{L}_{\text{train}}(\boldsymbol{\theta}_t)$ can cascade forward into a completely different $\boldsymbol{\theta_T}$, resulting in large fluctuations of the hypergradients. This noise makes the generalization loss hard to minimize (Eq 1). We find that the two main causes for this noise are the ordering of the training minibatches, and the weight initialization $\boldsymbol{\theta}_0$. Ideally, the hyperparameters we learn should be agnostic to both of these factors, and so we would like to average out their effect on hypergradients.

**Ensemble averaging** The most obvious way to address the above is to compute all hypergradients across several random seeds, where each seed corresponds to a different dataset ordering and weight initialization. We could then obtain an average hypergradient $\mu_t$ for each inner step $t$. Here, $\boldsymbol{\mu} \in \mathbb{R}^T$ is often called an ensemble average in statistical mechanics. The issue with ensemble averaging in our setting is its computational and memory cost, since each random seed requires differentiating through $T$ unrolled inner steps for $T$ hyperparameters. This makes both reverse-mode and forward-mode differentiation intractable. We consider the ensemble average as optimal in our analysis, which allows us to derive an expression for the mean square error between our hypergradients estimate and $\boldsymbol{\mu}$.

**Time averaging** In FDS, we use the long horizon to our advantage and propose to do time averaging instead of ensemble averaging, i.e. we *average out hypergradients across the inner loop (Eq 2) rather than the outer loop (Eq 1)*. More specifically, we sum hypergradients from $W$ neighbouring time steps in the inner loop, which is exactly equivalent to sharing one hyperparameter over all these steps. The average is then obtained trivially by dividing by $W$. For instance, when learning the learning rate schedule $\boldsymbol{\alpha}$ for $H = 10^4$ inner steps, we can learn $\boldsymbol{\alpha} \in \mathbb{R}^{10}$ where each learning rate is used for a window of $W = 10^3$ contiguous gradient steps. A stochastic system where the time average is equivalent to the ensemble average is called *ergodic* [50] and has been the subject of much research in thermodynamics [51] and finance [52]. In our case, hypergradients aren't generally ergodic, and so using a single average hypergradient for $W$ contiguous steps can introduce a bias. Informally, time averaging contiguous hypergradients leads to both noise reduction and bias increase, and we flesh out the nature of this trade-off in Theorem 4.1.

**Theorem 4.1.** Let each time step $t \in \{1, 2, ..., T\}$ have a corresponding hyperparameter $\lambda_t$ and non-greedy hypergradient $g_t = \partial L_{\text{val}}(\boldsymbol{\theta}_T)/\partial \lambda_t$. Each $g_t$ is viewed as a random variable due to the sampling process of the weight initialization $\boldsymbol{\theta}_0$ and the inner loop minibatch selection. Let $\boldsymbol{g} = [g_1, g_2, \dots, g_T]$ be sufficiently well approximated by a Gaussian distribution $\boldsymbol{g} \sim \mathcal{N}(\boldsymbol{\mu}, \boldsymbol{\Sigma})$, with mean $\boldsymbol{\mu} = [\mu_1, \mu_2, ..., \mu_T]$ and covariance matrix $\boldsymbol{\Sigma}$, where $\boldsymbol{\mu}$ corresponds to the optimal hypergradients. Assume that the changes in the values of $\boldsymbol{\mu}$ over time are bounded, i.e. $\boldsymbol{\mu}$ can be written as the $\epsilon$-Lipschitz function $\mu_{t+1} = \mu_t + \epsilon_t$, where $\epsilon_t \in [-\epsilon, \epsilon]$. Finally, let $c \in [0, 1]$ denote the maximum absolute correlation between the values of $\boldsymbol{g}$, i.e. $c \geq |\Sigma_{tt'}|/\sqrt{\Sigma_{tt}\Sigma_{t't'}}$ $\forall t \neq t'$. Then, we show that the mean squared error of the hypergradients with respect to $\boldsymbol{\mu}$ when sharing $W$ contiguous hyperparameters has an upper bound:

$$\text{MSE}_W \leq \frac{(1 + c(W - 1))}{W}\text{MSE}_1 + \epsilon^2 \frac{(W^2 - 1)}{12} \tag{5}$$

$$\text{where} \quad \text{MSE}_1 = \frac{1}{T}\sum_t \boldsymbol{\Sigma}_{tt} \tag{6}$$

and so for sufficiently small $\epsilon$ and $c$ we have with certainty, $MSE_W < MSE_1$ for some $W > 1$, where $MSE_1$ is the hypergradient error without any sharing (See Appendix B for a proof).

**Discussion** Theorem 4.1 demonstrates how sharing $W$ contiguous hyperparameters has two effects: 1) it reduces the hypergradient noise by a factor $\mathcal{O}(W/(1 + cW))$ due to the averaging of noisy hypergradients, and 2) it increases the error up to an amount $\mathcal{O}(\epsilon^2 W^2)$ due to an induced bias. Intuitively, averaging contiguous hypergradients maximally reduces noise when they aren't correlated ($c = 0$), and minimally increases bias when they are drawn from distributions of similar means ($\epsilon = 0$). In the simpler case where hypergradients are iid and have the same variance at each step, namely $\boldsymbol{g} \sim \mathcal{N}(\boldsymbol{\mu}, \sigma^2 \mathbf{1})$, the expressions above become $\text{MSE}_1 = \sigma^2$ and $\text{MSE}_W \leq \text{MSE}_1/W + \epsilon^2(W^2 - 1)/12$. Note that in all cases, the upper bound on $\text{MSE}_W$ has a single minimum $W^*$ corresponding to the optimal trade-off between noise reduction and bias increase.

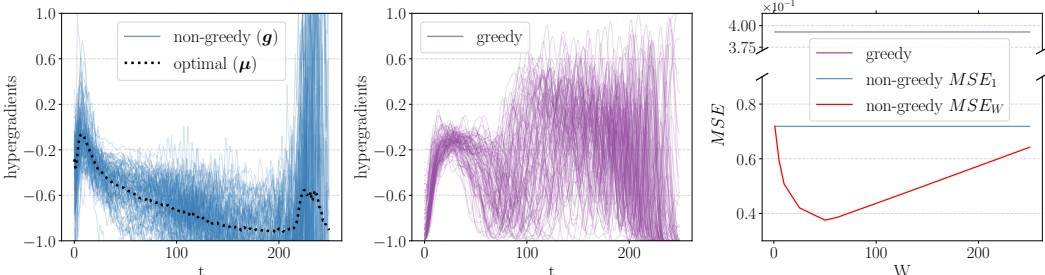

Figure 2: Hypergradients of $\boldsymbol{\alpha}$ on SVHN for 100 seeds in the non-greedy (left) and greedy (middle) setting. The mean squared error is also shown (right), and is calculated with respect to the optimal hypergradient ($\boldsymbol{\mu}$), i.e. the ensemble average of non-greedy hypergradients (dotted line in left figure). We can see that sharing hyperparameters over $W$ steps lowers the $MSE$ even for the small value of $T = 250$ used here. The best trade-off between noise reduction and bias increase is $W = 50$.

## 4.2 The FDS algorithm

As it is presented in Section 3.3, forward-mode differentiation would still have a memory cost that scales as $\mathcal{O}(DT)$ since we are learning one hyperparameter per step. However, addressing gradient degradation by sharing hyperparameters also conveniently reduces that memory cost by a factor $W$ down to $\mathcal{O}(DN)$, where $N = T/W$ is the number of unique hyperparameter values we learn. This is because we can safely average hypergradients without calculating them individually, by reusing the same column of $\boldsymbol{Z}$ for $W$ contiguous steps. This is shown in Algorithm 1, when learning a schedule of $N^\alpha$ learning rates with FDS. Here, $\boldsymbol{Z}^{\boldsymbol{\alpha}}_{[i]}$ refers to the i-th column of matrix $\boldsymbol{Z}^{\boldsymbol{\alpha}}$. We don't need to store $\boldsymbol{\mathcal{H}}$ or $\boldsymbol{A}^{\boldsymbol{\alpha}}$ in memory since we calculate the Hessian matrix product $\boldsymbol{\mathcal{H}}\boldsymbol{Z}^{\boldsymbol{\alpha}}$ directly. Most importantly, note that hyperparameters aren't updated greedily or online, but are updated once per outer step, which corresponds to differentiating through the entire unrolled inner loop optimization and getting the exact hypergradients $\partial\mathcal{L}_{\text{val}}(\boldsymbol{\theta}_T)/\partial\boldsymbol{\alpha}$.

The main cost of Algorithm 1 comes from calculating $\boldsymbol{\mathcal{H}}\boldsymbol{Z}^{\boldsymbol{\alpha}}$. There exists several methods to approximate this product, but we found them too crude for long horizons. This includes first-order approximations or truncation [32], which can be adapted to FDS trivially. One could also use functional forms for more complex schedules to be learned in terms of fewer hyperparameters, but this typically makes stronger assumptions about the shape of each hyperparameter schedule, which can easily cloud the true performance of HPO algorithms. In practice, we calculate $\boldsymbol{\mathcal{H}}\boldsymbol{Z}^{\boldsymbol{\alpha}}$ exactly, and use a similar form to Algorithm 1 to learn $\boldsymbol{\alpha}$, $\boldsymbol{\beta}$ and $\boldsymbol{\xi}$.

## 5 Experiments

Our experiments show how FDS mitigates gradient degradation and outperforms competing HPO methods for tasks with a long horizon. In Sections 5.1 and 5.2 we consider small datasets (MNIST and SVHN) and a small network (LeNet) to make reverse-mode differentiation tractable, so that the effect of hyperparameter sharing can be directly measured. In Sections 5.3 and 5.4, we then showcase FDS on CIFAR-10 where only forward-mode differentiation is

---

**Algorithm 1** Simplified FDS algorithm when learning $N^\alpha$ learning rates for the SGD optimizer with momentum. Each learning rate is shared over $W$ contiguous time steps

---

**Initialize:** $N^\alpha, W = T/N^\alpha, \boldsymbol{\alpha} = \boldsymbol{0}^{N^\alpha}$

*#outer loop*
**for** $o$ in $1, 2, ...$ **do**
  **Initialize:** $\mathcal{D}_{\text{train}}, \mathcal{D}_{\text{val}}, \boldsymbol{\theta}_0 \in \mathbb{R}^D,$
  $\boldsymbol{Z}^{\boldsymbol{\alpha}} = \boldsymbol{0}^{D \times N^\alpha}, C^{\boldsymbol{\alpha}} = \boldsymbol{0}^{D \times N^\alpha}$

  *#inner loop*
  **for** $t$ in $1, 2, ..., T$ **do**
    $\boldsymbol{x}_{\text{train}}, \boldsymbol{y}_{\text{train}} \sim \mathcal{D}_{\text{train}}$
    $\boldsymbol{g}_{\text{train}} = \partial\mathcal{L}_{\text{train}}(\boldsymbol{x}_{\text{train}}, \boldsymbol{y}_{\text{train}})/\partial\boldsymbol{\theta}$

    *#hyperparameter sharing*
    $i = \lceil t/W \rceil$
    $\boldsymbol{\mathcal{H}}\boldsymbol{Z}^{\boldsymbol{\alpha}}_{[1:i]} = \partial(\boldsymbol{g}_{\text{train}}\boldsymbol{Z}^{\boldsymbol{\alpha}}_{[1:i]})/\partial\boldsymbol{\theta}$
    $\boldsymbol{Z}^{\boldsymbol{\alpha}}_{[1:i]} = \boldsymbol{A}^{\boldsymbol{\alpha}}\boldsymbol{Z}^{\boldsymbol{\alpha}}_{[1:i]} + \boldsymbol{B}^{\boldsymbol{\alpha}}_{[1:i]}$

    update $C^{\boldsymbol{\alpha}}$ (Eq 4)
    $\boldsymbol{\theta}_{t+1} = \Phi(\boldsymbol{\theta}_t, \boldsymbol{g}_{\text{train}})$

  **end for**

  $\boldsymbol{g}_{\text{val}} = \partial\mathcal{L}_{\text{val}}(\mathcal{D}_{\text{val}})/\partial\boldsymbol{\theta}$
  $\boldsymbol{\alpha} \leftarrow \boldsymbol{\alpha} - 0.1 \times \boldsymbol{g}_{\text{val}}\boldsymbol{Z}^{\boldsymbol{\alpha}}/W$

**end for**

---

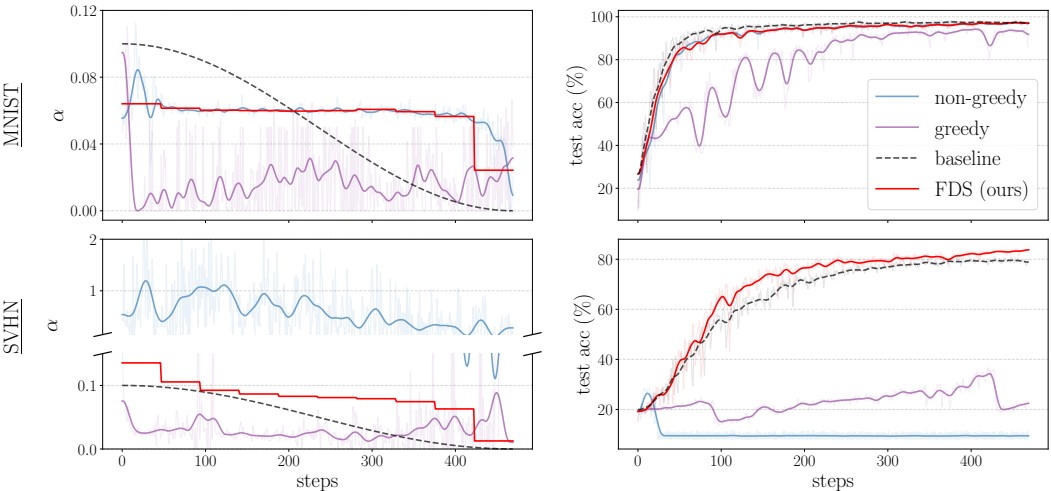

Figure 3: The learning rate schedule $\alpha$ learned for the MNIST and SVHN datasets using a LeNet architecture over one epoch. We observe that on real-world datasets like SVHN, both greedy and non-greedy hyperparameter optimizations fail to learn decent learning rate schedules when using one hypergradient per inner step. However, sharing learning rates over contiguous steps stabilizes non-greedy hypergradients and allows us to find learning rates that can even outperform reasonable off-the-shelf schedules in this setting.

tractable. All experiments are carried out on a single GTX 2080 GPU. More implementation details can be found in Appendix C.

## 5.1 The effect of hyperparameter sharing on hypergradient noise

We consider a LeNet network trained on an inner loop of $T = 250$ gradient steps, and calculate the hypergradients of the learning rate $\alpha$ across 100 seeds. Each seed is evaluated at the same value of $\alpha$, but corresponds to a different training dataset ordering and weight initialization. We can calculate the hypergradients greedily ($H = 1$) and non-greedily ($H = T$). The optimal hypergradients are considered to be the ensemble average of the non-greedy seeds as per Section 4.1, which allows an MSE to be calculated for each method. These results are shown in Figure 2 for a learning rate schedule initialized to small values. We observe that greediness is a poor approximation to the optimal hypergradients, and that time averaging contiguous hypergradients in the non-greedy case can significantly reduce the MSE even when averaging over only $W = 50$ steps.

The simplicity of this problem allows us to calculate $\Sigma$, $c$ and $\epsilon$ as defined in Theorem 4.1, to verify that our upper bound holds and that its shape as a function of $W$ is realistic. In the setting shown in Figure 2 we find $\epsilon = \max |\mu_{t+1} - \mu_t| \sim 0.08$ and $(1/T) \sum_t \Sigma_{tt} \sim 0.25$. The value of the maximum correlation between any two steps, $c$, can be quite high which makes the upper bound loose. In practice however, the shape of the upper bound in Theorem 4.1 as a function of $W$ (as plotted in Appendix B) closely matches that of the measured MSE shown in Figure 2. Note that the values of $\epsilon$, $\Sigma$, and $c$ can vary depending on the value of $\alpha$. As illustrated in Theorem 4.1, we find that settings that have a smaller values of $\epsilon_t$ benefit from a larger $W$ and reduce the MSE more (see Appendix D for more examples).

## 5.2 The effect of hyperparameter sharing on HPO

In the section above, we considered hypergradient noise around a fixed hyperparameter setting. In this section, we consider how that noise and its mitigation translate to meta-learning hyperparameters over several outer steps.

In Figure 3, we initialize $\alpha = 0$ and train a LeNet network over 1 epoch ($T \sim 500$ gradient steps) on MNIST and SVHN. We compare the maximally greedy setting ($H = 1$), the non-greedy setting ($H = T$), and FDS (also $H = T$ but with sharing of $W \sim 50$ contiguous hyperparameters). In

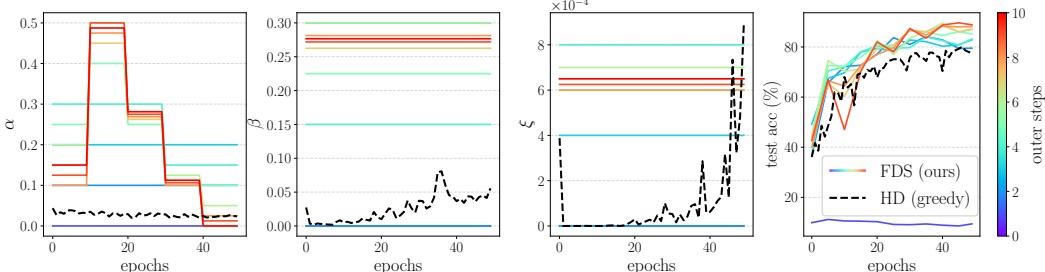

Figure 4: FDS applied to the SGD optimizer to learn (from left to right) the learning rate schedule $\alpha$, the momentum $\beta$, and weight decay $\xi$ for a WRN-16-1 on CIFAR-10. For each outer step (color) we solve CIFAR-10 from scratch for 50 epochs, and update all hyperparameters such that the final weights minimize some validation loss. We use hyperparameter sharing over $W = 10$, 50 and 50 epochs for $\alpha$, $\beta$ and $\xi$ respectively. All hyperparameters are initialized to zero and converge within just 10 outer steps to values that significantly outperform Hypergradient Descent (HD) [14], the greedy alternative. We match the performance of the best known hyperparameters in that setting and do so much faster than state-of-the-art black-box methods (see Section 5.4).

all cases we take 50 outer steps per hyperparameter. We make greediness more transparent by not using tricks as in Hypergradient Descent [14], where $\alpha_t$ is set to $\alpha_{t-1}$ before its hypergradients are calculated. As previously observed by [15], greedy optimization leads to poor solutions with learning rates that are always too small. While the non-greedy setting without sharing works well for simple datasets like MNIST, it fails for real-world datasets like SVHN, converging to much higher learning rates than reasonable. This is due to gradient degradation, whose negative effect can compound during outer optimization, as a single large learning rate can increase the hypergradient noise for neighbouring steps. As we share hyperparameters in FDS, we reduce and stabilize the outer optimization. This allows us to learn a much more sensible learning rate schedule, which can even outperform a reasonable cosine annealing off-the-shelf schedule on SVHN.

## 5.3 FDS on CIFAR-10

We demonstrate that our algorithm can be used to learn the learning rate, momentum and weight decay over 50 epochs of CIFAR-10 ($H \sim 10^4$) for a WideResNet of 16 layers (WRN-16-1). We choose not to use larger architectures or more epochs to enable compute time for more extensive comparisons and because hyperparameters matter most for fewer epochs. Note also that we are not interested in finding the architecture with the best performance, but rather in finding the best hyperparameters given an architecture. For the learning rate, we choose $W$ such that the ratio of $T/W$ is similar to the optimal one found in 5.1, and we set $W = T$ for the momentum and weight decay since only a single value is commonly used for these hyperparameters. The schedules learned are shown in Figure 4, which demonstrates that FDS converges in just 10 outer steps to hyperparameters that are very different to online greedy differentiation [14], and correspond to significantly better test accuracy performances. Note that having the maximum learning rate be large and occur half way during training is reminiscent of the one-cycle schedule [53].

## 5.4 FDS compared to other HPO methods

A common theme in meta-learning research has been the lack of appropriate baselines, with researchers often finding that random search (RS) can outperform complex search algorithms, for instance in NAS [54] or automatic augmentation [55]. In this section we compare FDS to several competing HPO methods on CIFAR-10, in both the performance of the hyperparameters found and the time it takes to find them. We consider Random Search (RS), Bayesian Optimization (BO), Hypergradient Descent (HD) [14], HyperBand (HB) [12] and the combination of BO and HB, namely BOHB [13]. The latter is typically regarded as state-of-the-art in HPO. We use the official HpBandster [56] implementation of these algorithms, except for HD which we re-implemented ourselves.

The performance of HPO methods is often clouded by very small search ranges. For instance, in [13] the learning rate is searched over the range $[10^{-6}, 10^{-1}]$. In the case of DARTS [38], expanding the search space to include many poor architectures has helped diagnose issues with its search algorithm [57]. For these reasons, we assume only weak priors and consider large search ranges: $\alpha \in [-1, 1]$, $\beta \in [-1.5, 1.5]$, and $\xi \in [-4 \times 10^{-3}, 4 \times 10^{-3}]$, which includes many poor hyperparameter values. In FDS we can specify search ranges by using a fixed update size $\gamma$ to the hyperparameters, which decays by 2 every time the hypergradient flips sign, which is common in sign based optimizers [58–61]. We found that black-box HPO methods did not scale well to more than $\sim 10$ hyperparameters for such large search ranges, and so considered learning 7 learning rates, 1 momentum and 1 weight decay over 50 epochs. Note that increasing these numbers doesn't affect the accuracy of FDS but significantly reduces that of RS, BO, HB and BOHB. The performance over time of the hyperparameters found by each method is shown in Figure 1. As is common in HPO, we plot regret (in test accuracy) with respect to the best hyperparameters known in this setting, which we obtained from an expensive grid search around the most common hyperparameters used in the literature (more details in Appendix F). To squeeze the optimal performance out of FDS in this experiment, we match the process used in HB and BOHB, namely using smaller budgets for some runs, in particular early ones. We can see that our method reaches zero regret in just 3 hours, while the next best method (BOHB) reaches zero regret in 60 hours. Other methods did not reach zero regret in the maximum of 100 hours they were run for. Note also that we retain the convergence speed of online greedy differentiation [14] while outperforming its regret by $\sim 10\%$ test accuracy.

## 6 Discussion

FDS is significantly better than modern HPO alternatives in the case of learning differentiable hyperparameters over long horizons. Its main advantage over black-box methods is its convergence speed, and its main advantage over other gradient-based methods is that it's non-greedy. Furthermore, forward-mode differentiation can provide exact hypergradients for any differentiable hyperparameter, contrary to some other approaches like implicit differentiation which approximates hypergradients, and can do so for certain types of differentiable hyperparameters only.

Nonetheless, it is worth pointing out the main limitations of FDS. First, black-box methods are slower but can readily tackle non-differentiable hyperparameters, while FDS would need to use relaxation techniques to differentiate through, say, discrete variables. Then, the computational and memory cost of FDS scales linearly with the number of hyperparameters. For instance, for a WideResNet of 16 layers we are limited in memory to $\sim 10^3$ hyperparameters on a single 12 GB GPU, which falls short of reverse-mode-based greedy methods which scale to millions of hyperparameters. Finally, while automatic forward-mode differentiation is becoming available for some toolboxes like JAX [62], it remains unavailable for the most popular deep learning libraries, which means that some pen-and-paper work is still required to derive hypergradients for given hyperparameters (Eq 4).

The focus of our future work will be on adapting FDS to the many meta-learning applications that rely on greedy optimization, such as differentiable architecture search, to improve their performance by making them non-greedy.

## 7 Conclusion

This work makes an important step towards gradient-based HPO for long horizons by introducing FDS, which enables non-greediness through the mitigation of gradient degradation. More specifically, we theorize and demonstrate that sharing hyperparameters over contiguous time steps is a simple yet efficient way to reduce the error in their hypergradients; a setting which naturally lands itself to forward-mode differentiation. Finally, we show that FDS outperform greedy gradient-based alternatives in the quality of hyperparameters found, while being significantly faster than all state-of-the-art black-box methods. We hope that our work encourages the community to reconsider gradient-based HPO in terms of non-greediness, and pave the way towards a universal hyperparameter solver.

**Acknowledgements**   The authors would like to thank Joseph Mellor, Antreas Antoniou, Miguel Jaques, Luke Darlow and Benjamin Rhodes for their useful feedback throughout this project. Our work was supported in part by the EPSRC Centre for Doctoral Training in Data Science, funded by the UK Engineering and Physical Sciences Research Council (grant EP/L016427/1) and the University of Edinburgh, as well as a Huawei DDMPLab Innovation Research Grant.

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
