# Appendices for:
# Gradient-based Hyperparameter Optimization Over Long Horizons

**Paul Micaelli**
University of Edinburgh
{paul.micaelli}@ed.ac.uk

**Amos Storkey**
University of Edinburgh
{a.storkey}@ed.ac.uk

## A: Forward-mode Hypergradient Derivations

Recall that we are interested in calculating

$$\boldsymbol{Z}_t = \boldsymbol{A}_t \boldsymbol{Z}_{t-1} + \boldsymbol{B}_t$$

recursively during the inner loop, where

$$\boldsymbol{Z}_t = \frac{d\boldsymbol{\theta}_t}{d\boldsymbol{\lambda}} \qquad \boldsymbol{A}_t = \frac{\partial \boldsymbol{\theta}_t}{\partial \boldsymbol{\theta}_{t-1}}\bigg|_{\boldsymbol{\lambda}} \qquad \boldsymbol{B}_t = \frac{\partial \boldsymbol{\theta}_t}{\partial \boldsymbol{\lambda}}\bigg|_{\boldsymbol{\theta}_{t-1}}$$

so that we can calculate the hypergradients on the final step using

$$\frac{d\mathcal{L}_{val}}{d\boldsymbol{\lambda}} = \frac{\partial \mathcal{L}_{val}}{\partial \boldsymbol{\theta}_T} \boldsymbol{Z}_T$$

Each type of hyperparameter needs its own matrix $\boldsymbol{Z}_t$, and therefore its own matrices $\boldsymbol{A}_t$, and $\boldsymbol{B}_t$. Consider first the derivation of these matrices for the learning rate, namely $\boldsymbol{\lambda} = \boldsymbol{\alpha}$. Recall that the update rule of SGD with momentum and weight decay after substituting the velocity $\boldsymbol{v}_t$ in is

$$\boldsymbol{\theta}_t = \boldsymbol{\theta}_{t-1} - \alpha_t \left( \beta_t \boldsymbol{v}_{t-1} + \frac{\partial \mathcal{L}_{train}}{\partial \boldsymbol{\theta}_{t-1}} + \xi_t \boldsymbol{\theta}_{t-1} \right)$$

and therefore it follows directly that

$$\boldsymbol{A}_t^{\boldsymbol{\alpha}} = \boldsymbol{1} - \alpha_t \left( \frac{\partial^2 \mathcal{L}_{train}}{\partial \boldsymbol{\theta}_{t-1}^2} + \xi_t \boldsymbol{1} \right)$$

The calculation of $\boldsymbol{B}_t^{\boldsymbol{\alpha}}$ is a bit more involved in our work because when using momentum, $\boldsymbol{v}_{t-1}$ is now itself a function of $\boldsymbol{\alpha}$. First we write

$$\boldsymbol{B}_t^{\boldsymbol{\alpha}} = -\beta_t \left( \frac{\partial \alpha_t}{\partial \boldsymbol{\alpha}} \boldsymbol{v}_{t-1} + \alpha_t \frac{\partial \boldsymbol{v}_{t-1}}{\partial \boldsymbol{\alpha}} \right) - \frac{\partial \alpha_t}{\partial \boldsymbol{\alpha}} \left( \frac{\partial \mathcal{L}_{train}}{\partial \boldsymbol{\theta}_{t-1}} + \xi_t \boldsymbol{\theta}_{t-1} \right)$$

$$= -\beta_t \alpha_t \frac{\partial \boldsymbol{v}_{t-1}}{\partial \boldsymbol{\alpha}} - \delta_t^{\otimes} \left( \beta_t \boldsymbol{v}_{t-1} + \frac{\partial \mathcal{L}_{train}}{\partial \boldsymbol{\theta}_{t-1}} + \xi_t \boldsymbol{\theta}_{t-1} \right)$$

35th Conference on Neural Information Processing Systems (NeurIPS 2021).

Now since

$$\boldsymbol{v}_t = \beta_t \boldsymbol{v}_{t-1} + \frac{\partial \mathcal{L}_{train}}{\boldsymbol{\theta}_{t-1}} + \xi_t \boldsymbol{\theta}_{t-1}$$

we can write the partial derivative of the velocity as an another recursive rule:

$$\begin{aligned}
\boldsymbol{C}_t^{\boldsymbol{\alpha}} &= \frac{\partial \boldsymbol{v}_t}{\partial \boldsymbol{\alpha}} \\
&= \beta_t \boldsymbol{C}_{t-1}^{\boldsymbol{\alpha}} + \frac{\partial^2 \mathcal{L}_{train}}{\partial \boldsymbol{\alpha} \partial \boldsymbol{\theta}_{t-1}} + \xi_t \frac{\partial \boldsymbol{\theta}_{t-1}}{\partial \boldsymbol{\alpha}} \\
&= \beta_t \boldsymbol{C}_{t-1}^{\boldsymbol{\alpha}} + \left( \xi_t \mathbf{1} + \frac{\partial^2 \mathcal{L}_{train}}{\partial \boldsymbol{\theta}_{t-1}^2} \right) \frac{\partial \boldsymbol{\theta}_{t-1}}{\partial \boldsymbol{\alpha}}
\end{aligned}$$

And putting all together recovers the system:

$$\begin{cases}
\boldsymbol{A}_t^{\boldsymbol{\alpha}} = \mathbf{1} - \alpha_t \left( \frac{\partial^2 \mathcal{L}_{train}}{\partial \boldsymbol{\theta}_{t-1}^2} + \xi_t \mathbf{1} \right) \\
\boldsymbol{B}_t^{\boldsymbol{\alpha}} = -\beta_t \alpha_t \boldsymbol{C}_{t-1}^{\boldsymbol{\alpha}} - \delta_t^{\otimes} \left( \beta_t \boldsymbol{v}_{t-1} + \frac{\partial \mathcal{L}_{train}}{\partial \boldsymbol{\theta}_{t-1}} + \xi_t \boldsymbol{\theta}_{t-1} \right) \\
\boldsymbol{C}_t^{\boldsymbol{\alpha}} = \beta_t \boldsymbol{C}_{t-1}^{\boldsymbol{\alpha}} + \left( \xi_t \mathbf{1} + \frac{\partial^2 \mathcal{L}_{train}}{\partial \boldsymbol{\theta}_{t-1}^2} \right) \boldsymbol{Z}_{t-1}^{\boldsymbol{\alpha}}
\end{cases}$$

For learning the momentum and weight decay, a very similar approach yields

$$\begin{cases}
\boldsymbol{A}_t^{\boldsymbol{\beta}} = \mathbf{1} - \alpha_t \left( \frac{\partial^2 \mathcal{L}_{train}}{\partial \boldsymbol{\theta}_{t-1}^2} + \xi_t \mathbf{1} \right) \\
\boldsymbol{B}_t^{\boldsymbol{\beta}} = -\beta_t \alpha_t \boldsymbol{C}_{t-1}^{\boldsymbol{\beta}} - \delta_t^{\otimes} (\alpha_t \boldsymbol{v}_{t-1}) \\
\boldsymbol{C}_t^{\boldsymbol{\beta}} = \delta_t^{\otimes} (\boldsymbol{v}_t) + \beta_t \boldsymbol{C}_{t-1}^{\boldsymbol{\beta}} + \left( \xi_t \mathbf{1} + \frac{\partial^2 \mathcal{L}_{train}}{\partial \boldsymbol{\theta}_{t-1}^2} \right) \boldsymbol{Z}_{t-1}^{\boldsymbol{\beta}}
\end{cases}$$

and

$$\begin{cases}
\boldsymbol{A}_t^{\boldsymbol{\xi}} = \mathbf{1} - \alpha_t \left( \frac{\partial^2 \mathcal{L}_{train}}{\partial \boldsymbol{\theta}_{t-1}^2} + \xi_t \mathbf{1} \right) \\
\boldsymbol{B}_t^{\boldsymbol{\xi}} = -\beta_t \alpha_t \boldsymbol{C}_{t-1}^{\boldsymbol{\xi}} - \delta_t^{\otimes} (\alpha_t \boldsymbol{\theta}_{t-1}) \\
\boldsymbol{C}_t^{\boldsymbol{\xi}} = \delta_t^{\otimes} (\boldsymbol{\theta}_{t-1}) + \beta_t \boldsymbol{C}_{t-1}^{\boldsymbol{\xi}} + \left( \xi_t \mathbf{1} + \frac{\partial^2 \mathcal{L}_{train}}{\partial \boldsymbol{\theta}_{t-1}^2} \right) \boldsymbol{Z}_{t-1}^{\boldsymbol{\xi}}
\end{cases}$$

# B: Theorem 4.1: Proof

**Preamble**   Consider that each time step $t \in \{1, 2, ..., T\}$ has a corresponding hyperparameter $\lambda_t$ and hypergradient $g_t = \partial L_{val}(\boldsymbol{\theta}_T)/\partial \lambda_t$. Each $g_t$ is viewed as a random variable due to the sampling process of the weight initialization $\boldsymbol{\theta}_0$ and the inner loop minibatch selection. Assume that $\boldsymbol{g} = [g_1, g_2, \ldots, g_T]$ is sufficiently well approximated by a Gaussian distribution, where $\boldsymbol{g} \sim \mathcal{N}(\boldsymbol{\mu}, \boldsymbol{\Sigma})$, with mean $\boldsymbol{\mu} = [\mu_1, \mu_2, ..., \mu_T]$ and covariance matrix $\boldsymbol{\Sigma}$. Assume that the values of $\boldsymbol{\mu}$ can be written as the $\epsilon$-Lipschitz function $\mu_{t+1} = \mu_t + \epsilon_t$, where $\epsilon_t \in [-\epsilon, \epsilon]$. Note that in general, the gradients at different time steps may be correlated. Let the magnitude of the correlation be bounded by $c \in [0, 1]$:

$$\frac{|\Sigma_{tt'}|}{\sqrt{\Sigma_{tt}\Sigma_{t't'}}} \leq c \qquad \forall \ t \neq t' \tag{1}$$

Let $W$ define the size of a non-overlapping window over which hypergradients are averaged. This produces $K$ windows, where each window $k \in \{1, 2, ..., K\}$ contains the time steps $S^{(k)}$ i.e. $S^{(1)} = \{1, 2, \ldots W\}$, $S^{(2)} = \{W + 1, W + 2, \ldots 2W\}$, etc. For simplicity of analysis [1] we assume the chosen window sizes are divisors of $T$ such that $K = T/W$. Sharing hyperparameters over $W$ contiguous time steps amounts to using the average hypergradient $\bar{g}^{(k)}$ for each step in that window, where

$$\bar{g}^{(k)} := \frac{1}{W} \sum_{t \in S^{(k)}} g_t \tag{2}$$

We can now consider the mean squared error across all time steps when averaging contiguous hypergradients in non-overlapping windows of size $W$:

$$\text{MSE}_W = \frac{1}{K} \sum_k \frac{1}{W} \sum_{t \in S^{(k)}} \mathbb{E}\left[\left(\bar{g}^{(k)} - \mu_t\right)^2\right] \tag{3}$$

where all expectations in our proof are over $\boldsymbol{g} \sim \mathcal{N}(\boldsymbol{\mu}, \boldsymbol{\Sigma})$. Note the case $\text{MSE}_1$ gives the standard case where no averaging occurs ($K = T$).

**Theorem**   Then

$$\text{MSE}_1 = \frac{1}{T} \sum_t \boldsymbol{\Sigma}_{tt} \tag{4}$$

$$\text{MSE}_W \leq \frac{(1 + c(W - 1))}{W}\text{MSE}_1 + \epsilon^2 \frac{(W^2 - 1)}{12} \tag{5}$$

**Proof**   The case for $\text{MSE}_1$ follows trivially from the definition of variance:

$$\text{MSE}_1 = \frac{1}{T} \sum_t \mathbb{E}[(g_t - \mu_t)^2] = \frac{1}{T} \sum_t \boldsymbol{\Sigma}_{tt} \tag{6}$$

and so the mean squared error is the average of the variances. We now focus on the $W > 1$ case. Consider a window enumerated by $k$, and the vector of gradients within that window $\boldsymbol{g}^{(k)} = (g_t | t \in S^{(k)})$. Under the Gaussian assumption, that vector is Gaussian distributed with covariance $\boldsymbol{\Sigma}^{(k)}$, which is a block of the covariance matrix $\boldsymbol{\Sigma}$ corresponding to the variables in $\boldsymbol{g}^{(k)}$. Let $\bar{\mu}^{(k)} = (1/W) \sum_{t \in S^{(k)}} \mu_t$ be the average of means in window $k$. We consider the mean squared error from window $k$ as :

---

[1]This assumption is unnecessary and can be relaxed but would result in a more cumbersome theorem statement, as the final window of size $< W$ would need to be considered.

$$\text{MSE}^{(k)} = \frac{1}{W} \sum_{t \in S^{(k)}} \mathbb{E}\left[\left(\bar{g}^{(k)} - \mu_t\right)^2\right] \tag{7}$$

$$= \frac{1}{W} \sum_{t \in S^{(k)}} \mathbb{E}\left[\left(\bar{g}^{(k)} - \bar{\mu}^{(k)}\right)^2 + \left(\mu_t - \bar{\mu}^{(k)}\right)^2 - 2\left(\bar{g}^{(k)} - \bar{\mu}^{(k)}\right)\left(\mu_t - \bar{\mu}^{(k)}\right)\right] \tag{8}$$

$$= \mathbb{E}\left[\left(\bar{g}^{(k)} - \bar{\mu}^{(k)}\right)^2\right] + \frac{1}{W} \sum_{t \in S^{(k)}} \left(\mu_t - \bar{\mu}^{(k)}\right)^2 \tag{9}$$

Now $\bar{g}^{(k)} = (1/W)\mathbf{1}^T \boldsymbol{g}^{(k)}$, and $\bar{\mu}^{(k)} = (1/W)\mathbf{1}^T \boldsymbol{\mu}^{(k)}$, and so $\bar{g}^{(k)} - \bar{\mu}^{(k)} = (1/W)\mathbf{1}^T(\boldsymbol{g}^{(k)} - \boldsymbol{\mu}^{(k)})$. Hence

$$\mathbb{E}\left[\left(\bar{g}^{(k)} - \bar{\mu}^{(k)}\right)^2\right] = \frac{1}{W^2}\mathbb{E}\left[\mathbf{1}^T\left(\boldsymbol{g}^{(k)} - \boldsymbol{\mu}^{(k)}\right)\left(\boldsymbol{g}^{(k)} - \boldsymbol{\mu}^{(k)}\right)^T\mathbf{1}\right] \tag{10}$$

$$= \frac{1}{W^2}\mathbf{1}^T\boldsymbol{\Sigma}^{(k)}\mathbf{1} \tag{11}$$

Now let $\boldsymbol{D}$ be the diagonal matrix of variances, i.e. $\boldsymbol{D}_{ii} = \boldsymbol{\Sigma}_{ii}^{(k)}$ $\forall i$ and $\boldsymbol{D}_{ij} = 0$ $\forall i \neq j$. We use the correlation bound (1), which can be written as $|\boldsymbol{\Sigma}_{ij}^{(k)}| < c\left[\boldsymbol{D}^{\frac{1}{2}}\mathbf{1}\mathbf{1}^T\boldsymbol{D}^{\frac{1}{2}}\right]_{ij}$ $\forall i \neq j$, and this allows us to write an upper bound for the expression above:

$$\mathbb{E}\left[(\bar{g}^{(k)} - \bar{\mu}^{(k)})^2\right] \leq \frac{1}{W^2}\mathbf{1}^T[(1-c)\boldsymbol{D} + c\boldsymbol{D}^{\frac{1}{2}}\mathbf{1}\mathbf{1}^T\boldsymbol{D}^{\frac{1}{2}}]\mathbf{1} \tag{12}$$

$$= \frac{(1-c)}{W^2}\sum_i \boldsymbol{D}_{ii} + c\left[\frac{1}{W}\mathbf{1}^T\boldsymbol{D}^{\frac{1}{2}}\mathbf{1}\right]\left[\frac{1}{W}\mathbf{1}^T\boldsymbol{D}^{\frac{1}{2}}\mathbf{1}\right] \tag{13}$$

$$= \frac{(1-c)}{W^2}\sum_i \boldsymbol{D}_{ii} + c\left[\frac{1}{W}\sum_i \sqrt{\boldsymbol{D}_{ii}}\right]^2 \tag{14}$$

$$= \frac{(1-c)}{W^2}\sum_i \boldsymbol{\Sigma}_{ii}^{(k)} + c\left[\frac{1}{W}\sum_i \sqrt{\boldsymbol{\Sigma}_{ii}^{(k)}}\right]^2 \tag{15}$$

This expression can be simplified further using Jensen's inequality for square roots:

$$\mathbb{E}\left[(\bar{g}_k - \bar{\mu}_k)^2\right] \leq \frac{(1-c)}{W^2}\sum_i \boldsymbol{\Sigma}_{ii}^{(k)} + \frac{cW}{W^2}\sum_i \boldsymbol{\Sigma}_{ii}^{(k)} \tag{16}$$

$$= \frac{1 + c(W-1)}{W^2}\sum_i \boldsymbol{\Sigma}_{ii}^{(k)} \tag{17}$$

Now we return to the second part of (9). This second term can be bounded using the Lipschitz constraints. In particular for window size $W$, the maximum error is given when there is maximum deviation from the mean, which occurs when $\mu_t = \mu_{t-1} + \epsilon$. If we write the first mean in window $k$ as $\mu_1^{(k)}$ we have $\mu_t = \mu_1^{(k)} + (t-1)\epsilon$ $\forall t \in S^{(k)}$ and in that case $\bar{\mu}^{(k)} = \frac{1}{W}(\mu_1^{(k)} + (\mu_1^{(k)} + \epsilon) + (\mu_1^{(k)} + 2\epsilon) + \ldots + (\mu_1^{(k)} + (W-1)\epsilon) = \mu_1^{(k)} + \frac{(W-1)\epsilon}{2}$ and so $\mu_t - \bar{\mu}_k = (t-1)\epsilon + \frac{(W-1)\epsilon}{2}$. Note that this quantity is the same for all windows $k$. We can use it to write an upper bound as follows:

$$\frac{1}{W} \sum_{t \in S^{(k)}} \left( \mu_t - \bar{\mu}^{(k)} \right)^2 \leq \frac{1}{W} \sum_{j=1}^{W} \epsilon^2 \left( (j-1) - \frac{W-1}{2} \right)^2 \tag{18}$$

$$= \frac{\epsilon^2}{W} \sum_{j=0}^{W-1} \left( j - \frac{W-1}{2} \right)^2 \tag{19}$$

$$= \frac{\epsilon^2}{W} \sum_{j=0}^{W-1} j^2 - (W-1)j + \frac{(W-1)^2}{4} \tag{20}$$

$$= \frac{\epsilon^2}{W} \left( \frac{W(W-1)(2W-1)}{6} - (W-1)\frac{W(W-1)}{2} + \frac{W(W-1)^2}{4} \right) \tag{21}$$

$$= \epsilon^2 \frac{(W^2-1)}{12} \tag{22}$$

Hence combining (17) and (22) together into (9) we have

$$\text{MSE}^{(k)} \leq \frac{1 + c(W-1)}{W^2} \sum_i \mathbf{\Sigma}_{ii}^{(k)} + \epsilon^2 \frac{(W^2-1)}{12} \tag{23}$$

and so incorporating it into (3) we get

$$\text{MSE}_W = \frac{1}{K} \sum_{k=1}^{K} \text{MSE}^{(k)} \tag{24}$$

$$\leq \frac{W}{T} \sum_{k=1}^{K} \left( \frac{(1 + c(W-1))}{W^2} \sum_i \mathbf{\Sigma}_{ii}^{(k)} + \epsilon^2 \frac{(W^2-1)}{12} \right) \tag{25}$$

$$= \frac{(1 + c(W-1))}{WT} \sum_i \mathbf{\Sigma}_{ii} + \epsilon^2 \frac{(W^2-1)}{12} \tag{26}$$

$$= \frac{(1 + c(W-1))}{W} \text{MSE}_1 + \epsilon^2 \frac{(W^2-1)}{12} \qquad \square$$

For sufficiently small $\epsilon$ and $c$ we have with certainty, $MSE_W < MSE_1$ for some $W > 1$.

**Discussion** We assume that means $\boldsymbol{\mu}$ can be written as the $\epsilon$-Lipschitz function $\mu_{t+1} = \mu_t + \epsilon_t$, where $\epsilon_t \in [-\epsilon, \epsilon]$. Generally speaking, contiguous hyperparameters have optimal values which are close, and therefore have close hypergradients during outer optimization. This assumption breaks if hyperparameters are initialized randomly, and so we initialize all of our hyperparameters to zero in our experiments. Ideally, we would solve for whole inner loop several times so that we can use the mean hypergradient $[\mu_0, \mu_1, ..., \mu_H]$ for each individual step, without doing any sharing. While we consider this to be the optimal hypergradients, this is too expensive in practice, and instead we consider averaging hypergradients from neighbouring inner steps. The result above indicates that when contiguous hypergradients are sufficiently de-correlated (small c), we can reduce the mean squared error by a factor $W$ compared to not averaging. However, if means $\mu_t$ drift over time by an amount $\epsilon_t \leq \epsilon$ this introduces some bias which increases the error and eventually results in $MSE_W > MSE_1$.

Finally, it is worth considering the simpler scenario when each hypergradient is considered to be drawn independently, i.e. $\boldsymbol{g} \sim \mathcal{N}(\boldsymbol{\mu}, \sigma^2 \mathbf{1})$. In that case, $c = 0$ and the mean squared errors become:

$$\text{MSE}_1 = \sigma^2 \tag{27}$$

$$\text{MSE}_W \leq \frac{\text{MSE}_1}{W} + \epsilon^2 \frac{(W^2-1)}{12} \tag{28}$$

**Visualizing the MSE for various $\mu_t$ profiles** . Since the mean squared error depends on the specific shape of $\mu_t$, we sample random $\mu_t$ profiles and show how their $MSE$ evolves as a function of $W$. This illustrates how tight the upper bound is.

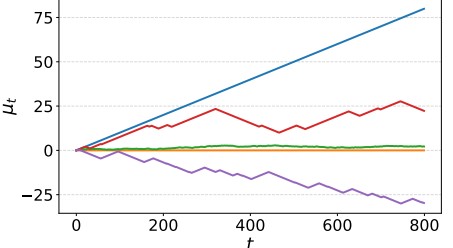 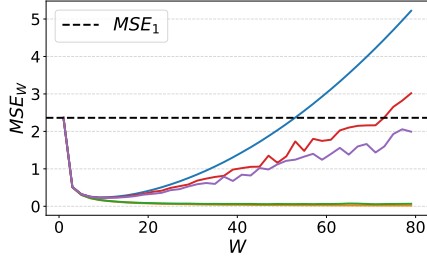

Figure 1: Several $\mu_t$ profiles and their corresponding mean squared error when sharing over $W$ contiguous steps, as a function of $W$. The blue and yellow curve correspond to the maximal and minimal drifts scenarios respectively.

# C: Implementation Details

We use a GeForce RTX 2080 Ti GPU for all experiments. We found that much of the literature on greedy methods uses the test set as the validation set, which creates a risk of meta-overfitting to the test set. Instead, we always carve out a validation set from our training set.

**Figure 1** Here we used very similar settings as Figure 4 for FDS, except we learned 7 learning rates to make the search space a bit more challenging. We use the HyperBanster HPO package for RS, BO, HB and BOHB. For HB and BOHB, the minimum budget argument is set to 1 epoch to allow for lots of fast evaluations, and the maximum budget is set to 50 epochs. We also use this technique to bring down our convergence time from $\sim 10$ hours to $\sim 3$ hours, namely we calculate hypergradients based on 10 epochs for some outer steps, rather than calculate all hypergradients on 50 epochs. Since HD needs the user to specify initial hyperparameter values, we random search over those for several consecutive HD runs.

**Figure 2** We calculate the hypergradient with respect to some learning rate schedule over 100 seeds, where each seed corresponds to a different training set ordering and network initialization. The learning rate schedule is fixed, and initialized to be a cosine decay over the full 250 batches, starting at 0.01. The batch size is set to 128, and 1000 fixed images are used for the validation data.

**Figure 3** Here we used a batch size of 128 for both datasets to allow 1 epoch worth of inner optimization in about 500 inner steps. Clipping was restricted to $\pm 3$ to show the effect of noisy hypergradients more clearly. Since MNIST and SVHN are cheap datasets to run on a LeNet architecture, we can afford 50 outer steps and early stopping based on validation accuracy. All learning rates were initialized to zero.

**Figure 4** We learn 5 values for the learning rates, 1 for the momentum and 1 for the weight decay, to make it comparable to the hyperparameters used in the literature for CIFAR-10. A batch size 256 is used, with $5\%$ of the training set of each epoch set aside for validation. We found larger validation sizes not to be helpful. Hypergradient descent uses hyperparameters initialized at zero as well, and trains all hyperparameters online with an SGD outer optimizer with learning rate 0.2 and $\pm 1$ clipping of the hypergradients. As described in appendix G, we used a sign based outer optimizer with adaptive step sizes rather than some hand-tuned outer learning rate schedule. We used initial values $\gamma_\alpha = 0.1, \gamma_\beta = 0.15$ and $\gamma_\xi = 4 \times 10^{-4}$ but the performance barely changed when these values were multiplied or divided by 2. Since we take 10 outer steps and initialize all hyperparameters at zero, this defines a search ranges: $\alpha \in [-1, 1]$, $\beta \in [-1.5, 1.5]$, and $\gamma \in [-4 \times 10^{-3}, 4 \times 10^{-3}]$. The Hessian matrix product is clipped to $\pm 10$ to prevent one batch from having a dominating contribution to hypergradients.

# D: Other hypergradient examples

Figure 2 depends on the value of $\alpha$ at which hypergradients are calculated. For some learning rate schedules, contiguous hypergradients are sampled from closer distribution ($\epsilon$ small) and so sharing over larger windows is beneficial, as shown in the figure below.

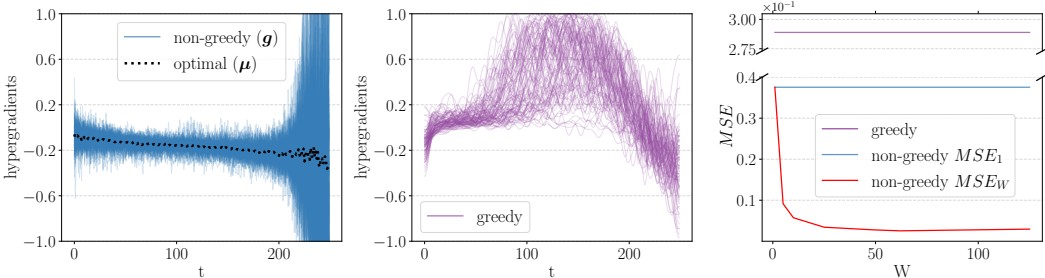

Figure 2: Similar to Figure 2 but for a smaller $\epsilon$. We can see that averaging hypergradients helps even more.

# E: Hypergradients

Here we provide the raw hypergradients corresponding to the outer optimization shown in Appendices: Figure 1. Note that the range of these hypergradients is made reasonable by the averaging of gradients coming from contiguous hyperparameters.

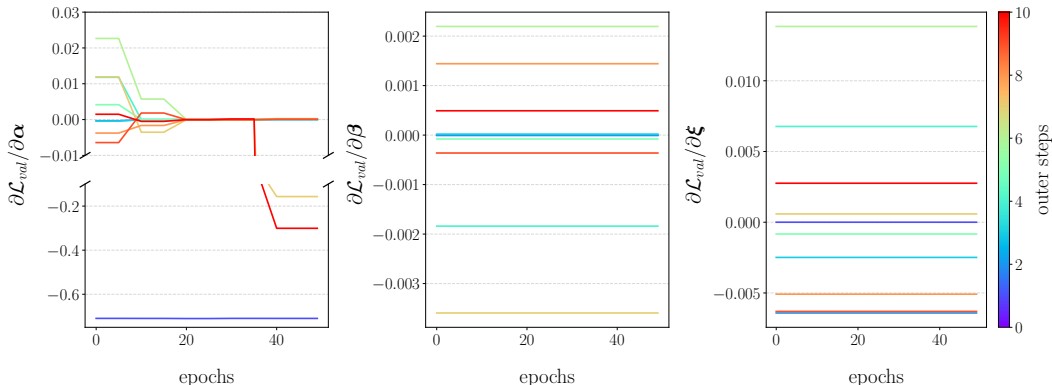

Figure 3: Hypergradients have a reasonable range but fail to always converge to zero when the validation performance stops improving.

# F: Baselines

The objective here is to select the best hyperparameter setting that a deep learning practitioner would reasonably be expected to use in our experimental setting, based on the hyperparameters used by the community for the datasets at hand. For CIFAR-10, the most common hyperparameter setting is the following: $\alpha$ is initialized at $\alpha_0 = 0.2$ (for batch size 256, as used in our experiments) and decayed by a factor $\eta = 0.2$ at $30\%, 60\%$ and $80\%$ of the run (MultiStep in Pytorch); the momentum $\beta$ is constant at 0.9, and the weight decay $\xi$ is constant at $5 \times 10^{-4}$. We search for combinations of hyperparameters around this setting. More specifically, we search over all combinations of $\alpha_0 = \{0.05, 0.1, 0.2, 0.4, 0.6\}$, $\eta = \{0.1, 0.2, 0.4\}$, $\beta = \{0.45, 0.9, 0.99\}$, and $\xi = \{2.5 \times 10^{-4}, 5 \times 10^{-4}, 1 \times 10^{-3}\}$. This makes up a total of 135 hyperparameter settings, which we each run 3 times to get a mean and standard deviation. The distribution of those means are provided in Figure 4, and the best hyperparameter setting is picked based on validation performance, which corresponds to $89.2 \pm 0.2\%$. Preliminary experiments showed that using schedules for the momentum and weight decay did not improve test accuracy.

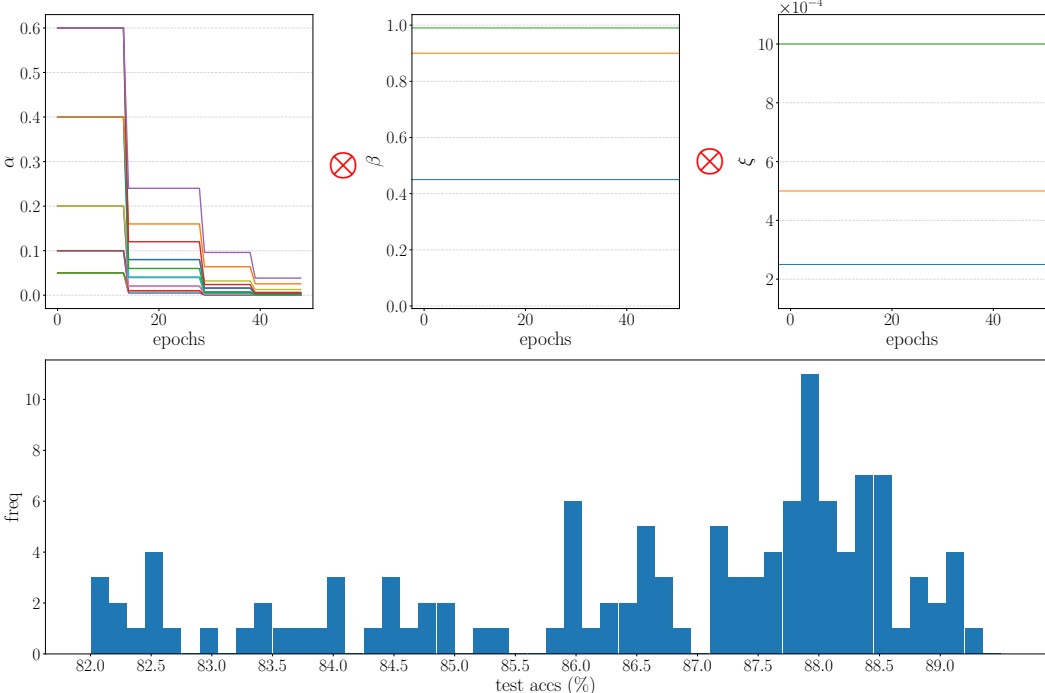

Figure 4: The combination of hyperparameters searched over for CIFAR-10 (top row) and the corresponding distribution of test accuracies (bottom row).

# G: Using Hypergradient Signs

While hyperparameter sharing produces stable hypergradients with a reasonable range (see Appendix E), tuning the outer learning rate schedule can be tedious and unintuitive, and doesn't allow the user to specify a range to search hyperparameters over. A simple and fairly common learning rate schedule consists in decaying the learning rate (e.g. by a factor of 2) every time the gradient changes sign [1]. The idea of using the sign of gradients to improve the efficiency of gradient descent dates back to the RPROP optimizer [2]. More recently, gradient signs have also been used to improve efficiency in the context of distributed learning [3], which has led to the discovery of robust convergence properties of sign-based SGD even in the case of biased gradients [4]. Using such a learning rate schedule for the outer optimizer frees us from having to tune the outer learning rate, but fails to define a search range for HPO. We can achieve this by updating each hyperparameter by an amount $\mathrm{sgn}(g) \times \gamma$, and letting $\gamma \leftarrow \gamma/2$ when the hypergradient $g$ changes sign across 2 consecutive outer steps. This allows for convergence after hypergradients have changed signs a few times. Being able to define the range of hyperparameter search more explicitly is especially useful to compare FDS to other HPO algorithms which also use a fixed search range (section 5.4).

---

**Algorithm 1** FDS algorithm similar to algorithm 1 but using hypergradient signs to update hyperparameters.

---

**Initialize:** $N^{\alpha}, W = T/N^{\alpha}, \boldsymbol{\alpha} = \mathbf{0}^{N^{\alpha}}$,
$\boldsymbol{\gamma} \in \mathbb{R}^{N^{\alpha}}$

*#outer loop*
**for** $o$ in $1, 2, ...$ **do**
    **Initialize:** $\mathcal{D}_{train}, \mathcal{D}_{val}, \boldsymbol{\theta}_0 \in \mathbb{R}^D$,
    $\boldsymbol{Z}^{\boldsymbol{\alpha}} = \mathbf{0}^{D \times N^{\alpha}}, C^{\boldsymbol{\alpha}} = \mathbf{0}^{D \times N^{\alpha}}$

    *#inner loop*
    **for** $t$ in $1, 2, ..., T$ **do**
        $\boldsymbol{x}_{train}, \boldsymbol{y}_{train} \sim \mathcal{D}_{train}$
        $\boldsymbol{g}_{train} = \partial \mathcal{L}_{train}(\boldsymbol{x}_{train}, \boldsymbol{y}_{train})/\partial \boldsymbol{\theta}$

        *#hyperparameter sharing*
        $i = \lceil t/W \rceil$
        $\mathcal{H} \boldsymbol{Z}^{\boldsymbol{\alpha}}_{[1:i]} = \partial(\boldsymbol{g}_{train} \boldsymbol{Z}^{\boldsymbol{\alpha}}_{[1:i]})/\partial \boldsymbol{\theta}$
        $\boldsymbol{Z}^{\boldsymbol{\alpha}}_{[1:i]} = \boldsymbol{A}^{\boldsymbol{\alpha}} \boldsymbol{Z}^{\boldsymbol{\alpha}}_{[1:i]} + \boldsymbol{B}^{\boldsymbol{\alpha}}_{[1:i]}$

        update $C^{\boldsymbol{\alpha}}$ (Eq 4)
        $\boldsymbol{\theta}_{t+1} = \Phi(\boldsymbol{\theta}_t, \boldsymbol{g}_{train})$

    **end for**

    $\boldsymbol{g}_{val} = \partial \mathcal{L}_{val}(\mathcal{D}_{val})/\partial \boldsymbol{\theta}$
    $\boldsymbol{s}_o = \mathrm{sgn}(\boldsymbol{g}_{val} \boldsymbol{Z}^{\boldsymbol{\alpha}}/W)$

    **for** n in $1, 2, ..., N_{\alpha}$ **do**
        **if** $\boldsymbol{s}_{o,[n]} \neq \boldsymbol{s}_{o-1,[n]}$ **then**
            $\boldsymbol{\gamma}^{\boldsymbol{\alpha}}_{[n]} \leftarrow \boldsymbol{\gamma}^{\boldsymbol{\alpha}}_{[n]}/2$
        **end if**
    **end for**

    $\boldsymbol{\alpha} \leftarrow \boldsymbol{\alpha} - \boldsymbol{s}_o \odot \boldsymbol{\gamma}^{\boldsymbol{\alpha}}$

**end for**

---