# OpenReview forum: "Gradient-based Hyperparameter Optimization Over Long Horizons"
_NeurIPS.cc/2021/Conference — NeurIPS 2021 Poster_

### Official Review · Reviewer_E74o · 2021-07-14

**Rating:** 6
**Confidence:** 3

**Summary:**

This paper tackles the problem of gradient-based optimization of hyperparameters over long training horizons, considering specifically optimizer hyperparams like learning rates (schedules), weight decay, and momentum, which are not easily optimized by scalable implicit differentation approaches.

The paper proposes to use  forward-mode (FM) differentiation to learn these hyperparameters, and contributes an algorithm to do so. FM differentiation  has the advantage that it scales better with memory over long horizons, unlike reverse-mode exact differentiation through the learning process.

In experiments, the paper  demonstrates that it does not suffer from the problems that greedy/online hyperparameter optimization approaches might face, and that it can be much more time efficient than gradient-free approaches like bayesian optimisation.



**Limitations And Societal Impact:**

Yes -- a good discussion of these.

**Main Review:**

The approach taken by the work is original: they combine work on forward mode differentiation with a nice strategy for reducing gradient degradation through the use of time-sharing hyperparameters. This naturally leads to the learning of e.g. learning rate schedules for optimizers.  They also provide a clear empirical study of the potential problems with greedy/online approaches for hyperparameter optimization.  The theoretical analysis in the paper of gaussian hypergradients and the associated reduction in MSE is clear, and is well-supported by Figure 2, right.

The paper has a clear algorithmic contribution and also good empirical results on MNIST, SVHN, and CIFAR-10. It is also written clearly and the results are well-presented. However, there are certain things that would be good to improve:

- The paper's abstract refers to general hyperparameters, whereas the paper itself mainly concerns itself with a small set of optimizer hyperparameters, and that too focuses on only SGD with momentum/weight decay. It would be good to clarify the scope in the abstract, and also include some further experiments on other optimizers at least (eg Adam), to demonstrate that improvements can still be obtained in other settings. If there are other low-dimensional hyperparameters that could be analysed, that would make the paper a lot stronger too.
- The results are mostly on three smaller-scale datasets; it would be good to at least include one or two other settings. For example, some similar results on CIFAR-100 or CUB-200 would make this paper stronger.
- Minor: error bars on Fig 1 might be good to get a sense of variation across seeds.
- Very minor: on line 193, is T hyperparameters a typo?

Overall, I think this is a good paper, but it would benefit from some further experimental study, so for now it is just below the acceptance threshold. If the authors can clarify the scope, and run  experiments on another optimizer/one other dataset, I am happy to raise my score.

# Update post author response
Thank you very much for your answers to my questions! The comments regarding testing on Adam and CIFAR-100 make sense -- if these could be included in the appendix of the final paper, that would be great. Similarly with the error bars -- I understand that it will make the figure harder to read, so perhaps an appendix version of this figure with the detail would be nice to include.  I am raising my score accordingly.

**Time Spent Reviewing:**

4

---

> ### Author Response · Authors · 2021-08-10
> **Reply to reviewer**
>
>
> We thank the reviewer for their review and encouraging comments. We hope to address the two main points raised below.
>
> ### Clearer scope
> While in principle the applicability of FDS is as large as that of forward-mode differentiation, in practice we have implemented it for a specific choice of hyperparameters, which have A and B matrices (Eqt 3) that would be different from A and B matrices of other hyperparameters. More specifically, the key insights behind what makes FDS work (Theorem 4.1) hold true for any differentiable hyperparameter. As forward-mode autodiff becomes popular (JAX has a beta version for it, which we deemed too immature for the research community at this stage), the code for FDS could be made entirely hyperparameter agnostic. For now, we have opted for a manual derivation of these A and B matrices for the hyperparameters at hand in our codebase, because it makes forward-mode more transparent for research. For instance, having access to A, B, C and Z (Eqt 3 & 4) allowed us to try conditioning/ablating on some of them, as well as truncating some of their update steps. We have made the abstract clearer by explicitly stating that we implement FDS for learning rate, momentum and weight decay.
>
> ### Other hyperparameters/datasets
>
> Broadly speaking, we have indeed gone for depth rather than breadth in our experiments. This is partly due to our lack of compute relative to the cost of “valid” HPO experiments (Figure 1 contains 8 seeds *6 methods *60 hours = 3000 GPU hours, all from a handful of GPUs).
>
> Prior to your comment, we had checked that FDS obtains zero regret on other hyperparameters like Adam and the mean/variance of the Gaussian initialization of a neural network weights (via the reparameterization trick). However we did not compute a Figure 1 equivalent for these settings because we found them less insightful, since the performance of the model was less affected by hyperparameters, and thus were essentially an easier task to solve (e.g. Adam internally regulates the effective learning rate over time, BN layers make initialization less relevant etc.).
>
> Since reading your comment, we have run FDS experiments on CIFAR-100 without changing the code. While similar schedules as that in Figure 4 can be obtained in a similar time, we also observe that FDS is less consistent across seeds for CIFAR-100 than CIFAR-10, and so would likely not obtain an average zero-regret (as boasted in Figure 1 on CIFAR-10). This is likely due to the CIFAR-100 validation loss being more ill-conditioned. Zero-regret average performance (over many seeds) for datasets like CIFAR-100 and ImageNet are thus a clear path of improvement for the field of gradient-based HPO, and a Figure 1 style CIFAR-100 experiment will be added in the appendix for that purpose. This also testifies to the sheer difficulty of differentiating through 10^4 gradient steps, and we believe our achievement on CIFAR-10/WRNs is still a significant step from the current state of the field.
>
> ### Minor points
>
> Line 193 isn’t a typo since ensemble averaging without any time averaging would indeed require one hypergradient per inner step across several seeds, and we have T inner steps in total. We have tried to put error bars in Figure 1 but some methods (RS and BO) have such high standard deviations that it makes the figure messy / harder to read.
>
> ### Conclusion
>
> We have clarified the scope in the abstract and provided evidence for broader experiments. While there is still much progress to be made in the field of gradient-based HPO, we believe that our current experiments do support our key claim/contribution: it is possible to differentiate through 10^4 optimization steps non-greedily in some settings, and by doing so get sota HPO results an order of magnitude faster. We believe that this alone could be very valuable for the community to build on, and we hope to have earned your recommendation for publication.

---

### Official Review · Reviewer_MsYr · 2021-07-17

**Rating:** 6
**Confidence:** 4

**Summary:**

Real-time recurrent learning (RTRL) is a useful approach for computing the gradient of hyperparameters through an inner optimization problem, because its memory usage does not scale with the length of the task. This paper introduces a method for hyperparameter optimization called Forward-Mode Differentiation with Sharing (FDS). The main contribution in FDS is the notion of sharing the same hyperparameter over a contiguous chunk of steps of the inner optimization, to achieve a kind of averaging over weights and minibatches. The paper discusses the trade-off between reducing noise and increasing bias when sharing hyperparameters over more contiguous steps: it provides an upper bound on the error of the approximate hypergradient when averaging across chunks of time, compared to the optimal hypergradient. The authors evaluate FDS on MNIST, SVHN, and CIFAR-10, using LeNet and WideResNet-16-1; they use FDS to tune the learning rate, momentum, and weight decay for SGD+momentum. They show some illustrative small-scale experiments looking at the approximate hypergradients obtained with different methods, and visualizing the adaptation of learning rates over outer optimization steps by their method. They also compare hyperparameter optimization performance of FDS against several popular black-box methods, including random search, BayesOpt, Hyperband, BOHB, and hypergradient descent.

**Limitations And Societal Impact:**

Yes, the authors have mostly addressed the limitations of their work. (E.g., they point out that FDS is not applicable to discrete hyperparameters (unless we use relaxations.)

**Main Review:**

Pros
----
* The paper is well-written; the method is described clearly.
* The discussion of related work is pretty comprehensive, covering relevant black-box and gradient-based hyperparameter optimization.
* It is nice that the proposed approach (FDS) is able to obtain useful hypergradients over many inner optimization steps for a fairly large real-world model and task, training a ResNet on CIFAR-10.
* The authors provide a theorem quantifying the error of the FDS-estimated hypergradient (using shared hyperparameters over continugous chunks of inner training) compared to the true hypergradient, based on the horizon length over which hyperparameters are shared.
* Figure 2 is a nice visualization of the optimal vs "greedy" and "non-greedy" hypergradients.
* The rightmost plot in Figure 2 is also very interesting as it demonstrates that the best choice of chunk size is an intermediate value, which makes sense in terms of balancing between reducing noise and increasing bias.
* Provides comparisons to real-world methods for hyperparameter optimization, including random search, BayesOpt, Hyperband, Bayesopt-Hyperband (BOHB), and hypergradient descent (HD). However, I have concerns regarding the search spaces for these comparisons.


Issues and Questions
--------------------
* L178-179: "can be applied to any differentiable hyperparameter, albeit with appropriate A_t and B_t matrices"
    - Do you have to define these matrices manually? Why is this the approach used in the paper? It should be possible to define a differentiable version of any optimizer (such as SGDm, RMSprop, or Adam), and use autodiff to compute gradients rather than manually specifying matrices for use in forward-mode for each potential optimizer.

* L360-362: "forward-mode doesn't come in an automatic-differentiation package yet, like reverse differentiation"
    - You can do forward-mode via reverse-on-reverse in any package that supports reverse-mode (as described here: https://j-towns.github.io/2017/06/12/A-new-trick.html). Also, libraries like JAX include explicit support for forward-mode differentiation. This goes together with my question above about why it is necessary to hand-derive the update rules for different hyperparameters?

* The method is strange in that depending on the horizon over which hyperparameters are shared, the learned schedule is more or less fine-grained (since each chunk effectively has its own hyperparameters). I think it makes the most sense to consider learning a fixed outer parameter for the entire inner training procedure: this fixed outer parameter can still be high-dimensional, and can parameterize a learning rate schedule (e.g., a linear schedule).

* L342: "we match the process used in HB and BOHB, namely using smaller budgets for some runs, in particular early ones"
    - It would be good to have more detail about how this was done: why were some runs of FDS stopped early? Did the runs diverge? Or was there a manual or automatic curriculum that progressively increased the inner problem length?

* In Figures 1, 3, and 4, it would also be useful to show the validation losses achieved by each method, since this is what is being optimized by FDS.
	- For clarity, what objective is being optimized by the black-box approaches in Figure 1, is it validation loss or validation accuracy?

* The search spaces for the learning rate and momentum coefficient in Section 5.4 are strange: it looks like the range for the LR is [-1, 1], which includes negative LRs? Coupled with the fact that there are 7 learning rates to learn, I would imagine that choosing any of the 7 LRs to be negative would substantially harm performance, making it hard for many of the methods (RS, BO, HB, and BOHB) to find good hyperparameters, since they would try he extremes of each range and frequently sample at least one negative LR, which would likely cause training to diverge.
    - Also, what were the initializations of the hyperparameters for FDS? From the algorithm, it looks like the initial hyperparameters are 0, which sounds good. But would FDS recover from starting with a random initialization in the range [-1, 1] for the LR or momentum?
    - I think it would be good to see the same experiment but with learning rates in a range like [0,1] or [0,10] for all methods.
    - Another question is how the hyperparameter ranges were parameterized (e.g., are the LRs searched over in log-space)?

* Does FDS use any mapping to constrain the values of different hyperparameters, such as exponential, sigmoid, or softplus transforms?

* Why do the experiments consider 7 learning rates but only 1 momentum and 1 weight decay value? Why not also try schedules for the momentum and weight decay?

* FDS is based on forward-mode differentiation, and is thus related to forward-mode hyperparameter optimization approaches such as RTHO [1] and MARTHE [2]. I think it would help to clarify the discussion of "greediness" of these approaches: I don't think greedy is necessarily the right terminology, since one of the main purposes of RTRL is to allow for online updates to the parameters of a system while remaining unbiased; of course, it is well-known that in order to obtain the true gradient of the parameters when using online updates, one would need to use vanishingly small step sizes. I think the effect of online updates in RTRL would be better referred to as "hysteresis" rather than "greediness." (When training RNNs, one of the main advantages of RTRL compared to TBPTT is the ability to make unbiased online updates; the RNN is never unrolled over a full sequence before making a parameter update.)
	- I also think it would be good to explicitly compare to RTHO and MARTHE as gradient-based approaches in the experiments.
	- How well would FDS work if you updated the hyperparameters after each horizon H rather than only once at the end of the full length T run?

* Since FDS is gradient-based, it would be good to show how it scales with the number of hyperparameters, for example per-layer learning rates. Did the authors try this? Or maybe this is too costly in terms of memory, since the RTRL Jacobian has dimension (# model params x # hyperparams), which is fine.

[1] Franceschi et al., "Forward and Reverse Gradient-Based Hyperparameter Optimization" ICML 2017.

[2] Donini et al., "MARTHE: Scheduling the Learning Rate Via Online Hypergradients" IJCAI 2020.


Minor
-----
* On Line 108, the notation is a bit confusing because $\lambda \in \mathbb{R}^T$ implies that there is only a 1D hyperparameter used at each step, when there can be a vector of hyperparameters at each step; it would be clearer to write $\lambda \in \mathbb{R}^{M \times T}$, there $M$ is the number of hyperparameters.
* L118: "waves" --> "waives"
* L130: I'm not sure "subspace" is the right terminology here, because this has a specific mathematical definition. Is this sentence really referring to a subspace?
* Use $\mathcal{L}\_{\text{train}}$ rather than $\mathcal{L}\_{train}$


Post-Rebuttal
------------------
I have read the all the reviews and the author responses. I thank the authors for their rebuttal. Overall, I still have some concerns about the search space for comparisons. Also, I think the discussion of greediness and RTRL should be improved, and the positioning of FDS should be made clearer.

The search space is still quite strange, because it is a bit extreme to consider a space of [-1, 1] for learning rates; if the goal is to make the space uninformative, then why not [-100, 100]? I have not seen any prior work that uses a search space that goes into negative learning rates. Also, not using a log-scaled parameterization for this search space makes it very improbable that any of the baselines will find a good LR (even more so if one tried to tune the LR for Adam, where the optimal LR would be somewhere around 1e-3 or 1e-4). Learning rates (and also weight decay coefficients) are typically represented in log-space because reasonable values often range over several orders of magnitude. One would expect that using log-space would even benefit FDS, since it would more easily fine-tune the learning rate.

I think the main contribution of this paper is an investigation into the effect of using different numbers of chunks over which to share hyperparameters in an inner optimization problem, when using forward-mode differentiation for hyperparameter optimization. That is, the paper looks at a slight modification of RTRL.

**Time Spent Reviewing:**

10

---

> ### Author Response · Authors · 2021-08-10
> **Reply to reviewer**
>
> We thank the reviewer for their review and recommending to accept our paper. We have seldom, if ever, received such a complete review and we are very thankful for it.
>
>
> ### Generalization and forward mode auto-diff
>
> You raise a very valid point. Our comment on the lack of forward-mode autodiff for deep learning practitioners was implicitly finger-pointing the frameworks used by the average reader to our paper (Pytorch/Tensorflow). It is correct that recent ML toolboxes like JAX boast forward-mode autodiff, and as these tools mature our method could be coded in a way that is more hyperparameter agnostic. Note that the key insight of our paper (time averaging hypergradients enables gradient-degradation-free differentiation of long horizons and leads to sota HPO) would still be valid in a forward mode autodiff implementation. We’ve made that clearer in the text.
>
> While FDS as a method isn’t restricted to any differentiable hyperparameter, we have made the choice to compute hypergradients manually in our code, mainly for reasons of transparency. Indeed, having access to matrices A, B, C and Z (Eqt 3 & 4) allowed us a myriad of side experiments (conditioning some of these matrices, ablating them, truncating some of their update steps etc.). Additionally, we worried that producing code which combines an immature problem (i.e. FDS is the first to boast differentiation over so many inner steps) with an immature toolbox (JAX still in beta, suffers from poor readability/debugging, requires complex meta-learning structures to be parsed into jit-able vs non jit-able etc.) would drive researchers away from building on top of our method.
>
> ### Parameterized schedules, search ranges, log space
>
> We have tried to address this point in lines 246-252 of the paper: "One could also use functional forms for more complex schedules to be learned in terms of fewer hyperparameters, but this typically makes stronger assumptions about the shape of each hyperparameter schedule, which can easily cloud the true performance of HPO algorithms."
>
> You are indeed correct that in practice, to maximize performance it would make sense to use as much domain knowledge as possible and use a parameterized schedule specific to each hyperparameter being learned. By avoiding this setting, we hoped to make FDS more transparent, and to make it clear that it is working because of time-averaging hypergradients rather than domain knowledge/ other tricks. This choice also made theorem 4.1 more intuitive when expressing the MSE as a variance + bias component.
>
> This philosophy also explains why we haven't searched the learning rate in log space, why we haven't constrained the hyperparameters (with sigmoid, softplus etc.), and why we've used very large search ranges compared to typical HPO papers. It is correct that the [-1, 1] range for the learning rate slows down the trial-and-error based black box methods, but we argue that this harder setting is more interesting/real-world-like than the small search ranges typically used, since knowing a small search range around the optimal hyperparameter values is a luxury that seldom occurs in practice, and is thus somewhat of a contrived HPO setting. As we run Figure 1 for increasingly smaller search ranges, all HPO methods converge to the same performance, and in particular FDS and BOHB become equally good around the [0, 0.1] range.
>
> Some combinations of bad hyperparameter values (like learning rate = 5) can indeed put FDS in a state that it struggles to recover from. In practice, we found that batch normalization helps FDS recover quickly from moderately bad hyperparameter values (e.g. learning rate = 1, -0.1).
>
> ### Smaller budgets for some runs
>
> This trick was borrowed from HB/BOHB and isn't a response to FDS diverging in any way, but rather the idea that when beginning the search (first few outer steps) we can afford coarse hypergradients that are still good enough to make progress. For instance when the learning rate is zero, we don't need to run 50 epochs in order to know we have to increase it. We are still being non-greedy because we keep H=T, but we make T temporarily smaller.  As the hyperparameters get closer to optimal, we do need to differentiate through (close to) the full T=50 epochs to squeeze out the most performance.
>
> ### Greediness in other methods.
>  RTHO (Franceschi2017) is only slightly related to RTRL (Williams1989) in the way it relies on a recursive update for the Jacobian, but certainly doesn't benefit from the guarantees of RTRL which, as you pointed out, assumes the RNN setting + that weights remain fixed throughout the entire trajectory. In our paper, we made no attempt to quantify greediness, but rather treated greediness as a True/False property of HPO algorithms. HD, RTHO and MARTHE are all greedy in the sense that they fail to "learn hyperparameters that minimize the final validation loss", but rather all minimize some function of intermediate validation losses. This is equivalent to learning the right running pace over a marathon by splitting the marathon into 100-meter sprints; at best it's suboptimal and at worst it's hopeless. In chaotic systems where the inner loop can follow such different trajectories, it's even less likely to get greedy hypergradients that approximate non-greedy ones. The term “hysteresis” also holds but it seems that it holds in both the non-greedy and greedy scenario, since the hypergradient at step t is affected by all steps < t and > t. Thus it would be less useful, unless we’re missing something?
>
>  While there are methodological differences between HD, RTHO and MARTHE, we found that much of their performance gaps comes down to "tricks" like tuning the initial hyperparameter values, using the right outer optimizer/outer learning rate etc. RTHO does make some effort to be less greedy (i.e. hyperparameters near the end of the inner loop are increasingly less greedy in the sense that they use Z_t which is increasingly close to Z_T as t approaches T). In Figure 1 preliminary RTHO runs also fail to reach zero regret with a similar behaviour as HD, and the RTHO curve will be added when we have run 8 seeds for 60 GPU hours. MARTHE is trickier to add to Figure 1. On one hand, the method itself suffers from methodological/experimental issues (see ICLR reject reviews) but additionally, it relies on meta-meta-parameters which are provided/tuned for the learning rate schedule only in their paper. It is correct that evolutionary algorithms are non-greedy and thus would be an interesting extra method to benchmark against. We haven’t found any commonly used or published method of the kind applied to HPO however. We have also observed that black-box methods scale poorly with the number of hyperparameters by training them on smaller W values (e.g. even BOHB cannot find a zero-regret solution when there are 10 inner learning rates to learn).
>
> ### Misc
> - in Figure 1, black box methods minimize validation loss rather than maximize validation accuracy, but we found no noticeable change in performance between the two approaches.
> - We considered learning a single momentum and weight decay in Figure 1 because learning schedules for these hyperparameters did not improve performance of any HPO method (ours + baselines), and it made easier to measure regret compared to a setting which, as is common, uses a single value for weight decay and momentum.
> - minor points / typos have been corrected, thank you.
>
> ### Conclusion
> We hope that we have answered your questions, and convinced you that the key point our paper demonstrates (i.e. it’s possible to differentiate through very long unrolled inner loops + this leads to sota HPO in some settings) would be valuable for the community to build on.

---

### Official Review · Reviewer_3Csr · 2021-07-17

**Rating:** 6
**Confidence:** 3

**Summary:**

This paper studies the problem of gradient based hyperparameter optimization.

The paper makes a distinction between "greedy" (truncated horizons) and "non-greedy" (full horizon) hypergradients. The problem with greedy methods is that they are biased, while non-greedy methods suffer from poor conditioning due to the long multiplication chains inherent in computing the hypergradient over many time steps (eq. 3). This poor conditioning means that with respect to variation in the inner problem (e.g. due to initialization and mini batch ordering), exact (full horizon) hypergradients will have high variance.

The paper proposes using forward mode autodiff to compute hypergradients, and choosing hyperparameters that are tied across multiple timesteps in order to reduce hypergradient variance.

The paper demonstrates the method by tuning learning rate schedules and momentum and weight decay hyperparameters on SVHN, MNIST, and CIFAR-10.

**Main Review:**

I thought the paper was fairly well written, but I have a number of questions about the method.

I think the problem of "gradient degradation" and how it induces variance in the hypergradient could be made clearer. As far as I can tell, the paper treats gradient degradation as the problem of exploding and vanishing gradients (poor conditioning), due to the long chain of multiplications that arises in the hypergradient computation. This poor conditioning subsequently causes two additional issues. First, trying to do gradient based hyperparameter optimization will be difficult due to poor conditioning of the hypergradients. This is an issue even if the inner problem is completely deterministic (e.g. fixed random seed). Second, when the inner problem has some stochasticity (in this paper, the stochasticity arises from random initialization and mini batch ordering of the inner problem), then the hypergradients will have high variance. Do the authors think of "gradient degradation" as purely the second point? That's kind of how that section is written (it equates degradation to high variance). But it seems to me like the poor conditioning itself is also a big problem (even when there is no variance in the hypergradient), and one that the proposed method does not address (since the proposed method is focused on how to compute the hypergradient, not what to do with it). Do the authors agree with this statement? If not, what am I missing?

The "time-averaging" approach proposed in this paper is motivated as a way of reducing the variance of the hypergradient. I was confused as to which hypergradient this is referring to. Over a window of size W, there will be W different hypergradients. Averaging these gives a new hypergradient, which is said to have lower variance - but lower than what? What is the other gradient estimate that is used for comparison for this statement? Is it each of the individual hypergradients in the window?

In Figure 3, the learned schedule quenches the learning rate right at the end of training. Is this some artifact of having a fixed number of steps in the inner loop while learning the hyperparameters? If you train for longer, does that schedule become suboptimal?

Also in Figure 3, how come the non-greedy curve does so poorly for SVHN? Is it because the non-greedy gradients are high variance? What happens if you decrease the outer-loop learning rate?

It would be nice if the paper provided some intuition or guidance for when and how to choose the window size W. Are there cases where it is better to use the full (non-greedy) hypergradient? Does this method only work when the optimal underlying hyperparameter (e.g. optimal learning rate schedule) varies smoothly over iteration, so that chunking the hyperparameters into windows is not too harmful?

Finally, it would be much more convincing if the paper would greatly expand the set of problems used to test the method. Does this work for training other hyperparameters, outside of optimizer hyperparameters for training convolutional neural nets?

Other minor questions:
- Did you try using a rolling window (instead of fixed windows) to average hypergradients?
- In Figure 2a, do you have intuition for why the particular curve looks the way it does (with non-monotonic behavior)? Why are there two humps in the curve, one early in training and one close to the end?


**Time Spent Reviewing:**

3

---

> ### Author Response · Authors · 2021-08-10
> **Reply to Reviewer**
>
> We thank reviewer 2 for their thorough review and relevant questions. We do our best to answer them below.
>
> ### Hypergradients of lower-variance, but lower than what?
>
> First note that the claims we make consider the $MSE$ (~ variance + bias^2) instead of just the variance, since low-variance-high-bias hypergradients wouldn’t help outer optimization. Specifically, the claim we make is that time-averaging contiguous hypergradients reduces the $MSE$. But reduces the $MSE$ compared to what? Compared to not time-averaging them, i.e using the untouched hypergradients for each step. In both cases, the $MSE$ is computed w.r.t the optimal hypergradients ($\mu$ in theorem 4.1), which correspond to the hypergradients you would get from ensemble-averaging over an infinite number of random seeds / small perturbations to the loss manifold.
>
> To give an example, consider an inner loop with 4 steps, with one hyperparameter per step, and with optimal hypergradients $\mu = [\mu_1, \mu_2, \mu_3, \mu_4]$ (which we don’t have access to). The hypergradients we calculate are $g_1 = [1, 3, 4, 6]$. Now let us average over $W=2$ to get $g_2 = [2, 2, 5, 5]$. The claim of theorem 4.1 is that we can expect $(\mu - g_2)^2 < (\mu - g_1)^2$ i.e. $MSE_2 < MSE_1$. Informally, this is because (under some mild conditions) doing this time-averaging reduces the variance component of the $MSE$ by a lot, and only increases the bias component by a little.
>
> ### Poor conditioning and gradient degradation
>
> This is an excellent point. You are correct in stating that we approached the problem from the perspective of the second component to gradient degradation: reducing MSE/variance wrt to the ensemble average you would get when averaging over an infinite number of random seeds. In the case when there is no randomness to the inner loop, any hypergradient sample would just be equal to the ensemble average (as defined in the paper), and yet this sample might not be enough to optimize the validation loss well if the loss manifold itself is ill-conditioned/high curvature.
>
> One way to solve this dichotomy is to introduce our method into slightly more general terms. Rather than letting the optimal hypergradient be the ensemble average over different random seeds, we can let the optimal hypergradients be the ensemble average over small perturbations to the inner loop (note that a deterministic inner loop can still be perturbed). This corresponds to hypergradients calculated on some smoothed validation loss manifold (namely the average manifold over lots of slightly perturbed manifolds), which is easier to optimize. In this paradigm, it is still true that our time-average hypergradients would be a closer estimate to these optimal hypergradients, and mitigate ill-conditioning.
>
> Please let us know if we misunderstood part of your comment.
>
> ### Figure 3: quenching learning rate + bad non-greedy curve
>
> Learning a schedule that decays to small values at the end of the inner loop is indeed a consequence of (and a desired effect) any inner loop with a finite number of steps. In the greedy case, there is no awareness of inner time (i.e. each hyperparameter at time step t is optimized as if step t is the last step of the inner loop) and thus the learning rate doesn’t “globally” decay. For longer inner loops, the schedule learned on small inner loops would become increasingly sub-optimal, since this essentially introduces some domain shift for the meta parameters.
>
> The non-greedy curve for SVHN in Figure 3 does indeed fail due to the high variance of hypergradients and how that compounds over outer steps. Essentially, the high variance eventually causes one learning rate to be high, which makes the inner loop even more chaotic and increases the variance at other time steps, and so more learning rates become high etc. In the limit of infinitely small outer learning rates and infinitely many outer steps, this problem should go away. In practice however, too many outer steps lead to an undesirable compute cost.
>
> ### Optimal window size, conditions for FDS to help
>
> Note that the expression we derived in Equation 5 can be used to obtain an expression for the lower bound of the optimal window size, but it’s expressed in terms of $c$ and $\epsilon$ which we only have access to for toy problems. However, equation 5 does tell us that the $MSE$ as a function of $W$ has a single optimum irrespective of the validation loss manifold, which makes it easy to find in practice by trying a few values. Indeed Figure 2c does show a single minimum.
> You are correct to point out that FDS works best when the optimal underlying schedule is smooth. In the case where the optimal hyperparameter schedule varies abruptly (high $\epsilon$) but the variance of hypergradients is low (low $MSE_1$) then it’s possible for the time-averaged hypergradients to be biased-dominated and have $MSE_W > MSE_1$ (see theorem 4.1).
>
> ### Other hyperparameters
>
> It is correct that we have gone for depth rather than breadth in our experiments. We have checked that FDS obtains zero regret on other things like the Adam or the mean/variance of the initialization of a neural network weights (via the reparameterization trick). However we found these settings to be less insightful since the performance of the model was less affected by hyperparameters, and thus were essentially an easier task to solve (e.g. Adam internally scales the learning rate, BN layers make initialization less relevant etc.). Since an in-depth experiment in HPO can be very expensive in compute (Figure 1 contains 8 seeds *6 methods *60 hours = 3000 GPU hours) and we only have access to a handful of GPUs, we’ve not produced a similar figure for these hyperparameters. However, we believe that the main claim of the paper is sufficiently well supported by our sota results on the three SGD hyperparameters we considered (along with Theorem 4.1).
>
> ### Minor points
> Using a rolling window is a sensible idea that works well in reverse-mode differentiation but unfortunately is not amenable to forward-mode differentiation in a way that scales well in memory (i.e. in a way that allows matrix $Z$ to be of size $D \times (T/W)$). The bumps in figure 2a depend on the current schedule that is being used and aren’t static during outer optimization, but we do systematically observe more variance for the hypergradients of the last few steps, most likely because they are the ones to change the final weights the most in this setting.
>
> ### Conclusion
>
> We hope that our answers have earned your recommendation for publication, and we will actively answer any further points you may raise. We believe that the key claim of the paper (it is possible to differentiate through 10^4 optimization steps non-greedily in some settings, and by doing so get sota HPO results an order of magnitude faster) could be very valuable to build on for the community.

---

> > ### Author Response · Authors · 2021-08-24
> > **further questions**
> >
> > Thanks again for your time reviewing our paper. We hope that we were able to answer all your questions above. If not, and since your score is in contrast with the other three reviewers, we remain at your disposal for any further inquiry you may have. Thank you!

---

> > > ### Comment · Reviewer_3Csr · 2021-08-30
> > > **Thank you for your response**
> > >
> > > Hi,
> > >
> > > Thank you for your response (which was well written, and much appreciated). You have addressed my main concerns.
> > >
> > > One suggestion: your comment that the method induces a bias-variance trade-off (greatly reducing the variance for a mild increase in bias) was helpful, perhaps it would be useful to also emphasize that in the manuscript.

---

### Official Review · Reviewer_7zoB · 2021-08-01

**Rating:** 6
**Confidence:** 2

**Summary:**

The paper proposes a gradient based technique for efficient long horizon hyper-parameter optimization. The problem is quite important and the paper discusses the existing techniques and their shortcomings quite well.

The authors propose a technique which utilises forward-mode Automatic differentiation (AD). This helps overcome the gradient degradation which occurs in greedy gradient based hyper-parameter optimization methods like Hyper-gradient descent.



**Limitations And Societal Impact:**

The authors have discussed this quite well.

**Main Review:**

- Hyper-parameter optimization over long gradients is an important problem. The authors exploit forward mode differentiation to design a memory and compute efficient algorithm for the problem. The idea is good and authors show its effectiveness with good experimentation.

- The theoretical results and the discussions presented also back the hypothesis.

- The authors conduct thorough experiments and compare with a wide range of state-of-the-art baselines.

- The writing is also good. A figure showing a pictorial representation for the parameter sharing across time and forward-mode AD could be helpful.

- So PyTorch does not have forward mode differentiation. A lack of forward mode differentiation in common autodiff packages might make the adoption of this method a bit difficult. But I think this is a step in the right direction and it would be helpful for the community.

- But can you check if JAX or AutoGrad has forward mode differentiation implemented?

===========================================================
#### POST REBUTTAL

Thanks to the authors for their response. After going through the discussion, I have decided to retain my score.

**Time Spent Reviewing:**

5

---

> ### Author Response · Authors · 2021-08-10
> **Reply to Reviewer**
>
> We thank the reviewer for their review and recommending accepting our paper.
>
> You raise a very valid point. Our comment on the lack of forward-mode autodiff for deep learning practitioners was implicitly finger-pointing the frameworks used by the average reader to our paper (Pytorch/Tensorflow). It is correct that recent ML toolboxes like JAX boast forward-mode autodiff, and as these tools mature our method could be coded in a way that is more hyperparameter agnostic. Note that the key insight of our paper (time averaging hypergradients enables gradient-degradation-free differentiation of long horizons and leads to sota HPO) would still be valid in a forward mode autodiff implementation. We’ve made that clearer in the text.
>
> While FDS as a method isn’t restricted to any differentiable hyperparameter, we have made the choice to compute hypergradients manually in our code, mainly for reasons of transparency. Indeed, having access to matrices A, B, C and Z (Eqt 3 & 4) allowed us (and future researchers) several side experiments (conditioning/ablating some of these matrices, truncating some of their update steps etc.). Additionally, we worried that producing code which combines an immature problem (i.e. FDS is the first to boast differentiation over so many inner steps) with an immature toolbox (JAX still in beta, suffers from poor readability/debugging, requires complex meta-learning structures to be parsed into jit-able vs non jit-able etc.) would drive researchers away from building on top of our method.
>
> We hope that this answers your questions and remain available for further discussion.

---

### Decision · Program_Chairs · 2021-09-28

**Decision:**

Accept (Poster)

**Comment:**

All reviewers agree that this paper proposed a useful method. Some questions regarding clarity, implementation and experiments were raised but addressed by the authors in the rebuttal phrase. Some reviewers have made suggestions to improve clarity and presentation of the results which I urge the authors to take into account for the camera ready version.

**Consistency Experiment:**

NeurIPS has a long history of experimentation. In 2014, NeurIPS ran an experiment in which 10% of submissions were reviewed by two independent committees to quantify the randomness in the review process. This year, we repeated a variant of this experiment to see how the quality of the review process has changed over time.  This paper was part of the experiment and was therefore assigned to two committees (consisting of reviewers, an Area Chair, and a Senior Area Chair) that reached independent decisions.  If both committees made the same recommendation, this recommendation was followed. If a single committee recommended acceptance, the paper was accepted (with the exception of a few cases in which the other committee identified what we considered a fatal flaw, e.g., an error in a key result).

Both committees reached the same decision: **Accept (Poster)**

The other committee assigned to the paper recommended **Accept (Poster)**.  You can find the other set of reviews, along with any follow up discussion with the authors here:
https://openreview.net/forum?id=5lko4-9cRAk